# Role of CaMKIIa reticular neurons of caudal medulla in control of posture
Pavel V. Zelenin ⬤ , Vladimir F. Lyalka ⬤ , Shih-Hsin Chang ⬤ , Francois Lallemend ⬤ ,
Tatiana G. Deliagina & Li-Ju Hsu ⬤ ✉

Terrestrial quadrupeds stabilize dorsal-side-up body orientation (the vertical orientation of its dorso-ventral axis) through the postural control system, with supraspinal inputs, including those from the reticular formation, playing a central role. The contribution of specific molecularly identified reticular neuron populations to posture, however, has remained unclear. We investigated CaMKIIa-expressing reticular neurons (CaMKIIa-RNs) in the caudal medulla and their role in postural regulation. Using chemogenetic activation and inactivation in mice, we found that unilateral activation of CaMKIIa-RNs produced ipsilateral roll tilt of the head and trunk, driven by flexion/adduction of ipsilateral limbs and extension/abduction of contralateral limbs. This tilt was actively stabilized on a tilting platform and maintained during locomotion. In contrast, unilateral inactivation evoked opposite effects. Histological analysis revealed that CaMKIIa-RNs include reticulospinal neurons projecting via the ipsilateral lateral funiculus to the intermediate gray matter of the spinal cord. While both excitatory and inhibitory neurons are present, excitatory neurons predominate. Our results demonstrate that CaMKIIa-RNs in the caudal medulla are essential for maintaining dorsal-side-up orientation in diverse environments. Their left/right symmetry supports stability on horizontal surfaces, whereas asymmetry enables compensation on inclined surfaces, underscoring their key role in supraspinal control of posture.

The maintenance of the basic body posture—upright in humans and a dorsal-side-up (i.e., a vertical orientation of the body dorso-ventral axis) in terrestrial quadrupeds—is a vital motor function. Any deviation from this orientation induced by external forces triggers an automatic postural response—a corrective movement—aimed at restoring the initial orientation. Also, both humans and animals can specifically change the body configuration in context of different motor behaviors.

The basic postural networks reside in the brainstem, cerebellum, and spinal cord[1]. Supraspinal networks play a crucial role in control of posture. Distortions in activity of descending systems forming the output of the postural networks—potentially caused by spinal cord injury or by neurological diseases—results in severe postural dysfunctions[2–8].

It was suggested that there are two types of posture-related supraspinal influences: first, phasic postural commands[9–12] contributing to generation of postural corrections and, second, tonic drive activating spinal postural networks[4,13,14].

Neurons of the reticular formation of the brainstem are important elements of supraspinal postural networks. They form a number of reticular nuclei from which originates the phylogenetically oldest descending system, the reticulospinal one. In the lamprey—a lower vertebrate, the reticulospinal system is the only developed supraspinal system and it plays a key role in

control of posture. It was demonstrated that any deviation from the stabilized dorsal side up body orientation in the lamprey leads to activation of a specific population of reticulospinal neurons[15]. Each of the neurons in this population activates a specific motor synergy and collectively, the activated reticulospinal neurons evoke the motor output necessary for the postural correction[16]. Also, it was demonstrated that left/right asymmetry in activity of reticulospinal neurons shifts the set-point of postural control system leading to stabilization of a new orientation of the body in space[15].

A number of evidence indicates that reticulospinal neurons in higher vertebrate, including terrestrial quadrupeds, play similar functional roles in control of posture. It was demonstrated in cats that reticulospinal neurons in the pontomedullary reticular formation transmit phasic postural commands for generation of postural corrections caused by an unexpected drop of support under one of the limbs[11]. Previously, we demonstrated that in rabbits, binaural galvanic vestibular stimulation (GVS) evokes lateral body sway that is actively stabilized indicating that GVS shifts the set-point of the postural control system[17,18]. Since GVS evokes strong asymmetry in activity of vestibular afferents[19,20] and reticulospinal neurons receive substantial vestibular input[21,22], one can suggest that left/right asymmetry in activity of reticulospinal neurons contributes to the shift of the set-point of the postural system. It was also demonstrated that microstimulation of specific sites

Department of Neuroscience, Karolinska Institutet, Stockholm, Sweden. ✉e-mail: liju.hsu@ki.se

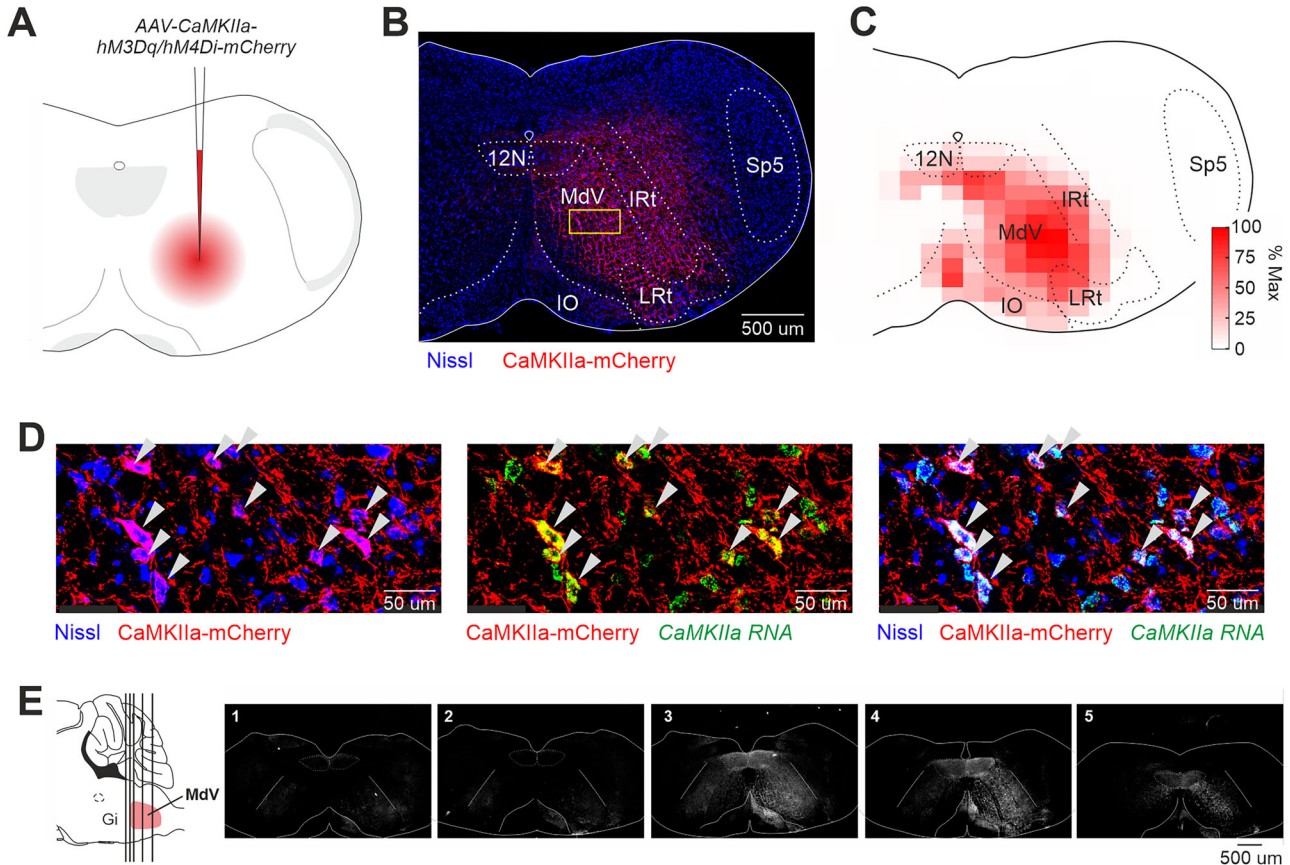

**Fig. 1 | Unilateral expression of DREADDS in CaMKIIa-RNs of the caudal medulla.** **A** A schematic drawing of the AAV-CaMKIIa-hM3Dq/hM4Di-mCherry injection site. **B** A representative example of unilateral infection of CaMKIIa-RNs with AAV-CaMKIIa-hM3Dq-mCherry. A confocal image illustrates mCherry-labeled CaMKIIa-expressing neurons in reticular formation of caudal medulla. The yellow rectangle highlights the region shown with higher magnification in panel (**D**). **C** A heatmap showing the average extent of the infected area ($N = 13$). **D** Combination of Nissl staining with immunochemistry for mCherry and in situ hybridization for *CaMKIIa* mRNA showing that the infected cells are CaMKIIa⁺ neurons (indicated by white arrowheads) in MdV. **E** Coronal sections from a representative animal illustrating the rostro-caudal distribution of mCherry-labeled CaMKIIa-expressing neurons following unilateral injection of the virus into the medullary reticular formation. Numbers 1–5 correspond to rostro-caudal levels shown on the scheme of the sagittal section of the brainstem (the left panel). MdV the medullary reticular nucleus ventral part, IRt the intermediate reticular nucleus, LRt the lateral reticular nucleus, IO the inferior olive: 12N the hypoglossal nucleus, Sp5 the spinal trigeminal nucleus.

within the pontomedullary reticular formation activates different motor synergies (involving muscles of both left and right limbs), which can contribute to generation of postural corrections as well as to changes of the body configuration in context of specific motor behaviors while stimulation of other sites evokes bilateral augmentation or suppression of the muscle tone[23].

Although it is documented that neurons of the pontomedullary reticular formation contribute to control of posture[24], the role of different molecularly identified reticular neurons located in different parts of reticular formation in control of particular aspects of posture, such as the body orientation in space, efficacy of postural corrections, specific changes of the body configuration, remains unknown.

Recent advances in genetics have inspired numerous studies striving to identify cell-type-specific functional roles in the control of movements, notably in the reticular nuclei. It was demonstrated that Vglut2 reticular neurons located in the lateral paragigantocellular nucleus are implicated in initiation of locomotion[25,26], while *Chx10* V2a neurons located in the gigantocellular reticular nucleus cause locomotor stop[27]. Also, it was shown that left/right asymmetry in the *Chx10* V2a activity evokes lateral turn[28,29].

Neurons expressing calcium/calmodulin-dependent protein kinase II alpha (CaMKIIa) were extensively studied in the context of synaptic plasticity, learning, and recovery after injury[30–32]. Although most work on CaMKIIa were focused on forebrain circuits, emerging evidence indicates that CaMKIIa expression is also present in defined populations within the reticular formation[33]. This raises the intriguing possibility that CaMKIIa-expressing reticular neurons may control specific aspects of motor behavior and thus, potentially, CaMKIIa could serve as a molecular marker to access specific functional neuronal populations. In the present study, we revealed a specific population of reticular neurons, the CaMKIIa expressing reticular neurons (CaMKIIa-RNs), located in the caudal medulla that control the body orientation in the transverse plane in mice.

## Results

To investigate the role of CaMKIIa-RNs in control of posture, we used a chemogenetic approach. First, excitatory (hM3Dq) or inhibitory (hM4Di) DREADDs were expressed unilaterally in CaMKIIa-RNs of the caudal medulla in mice (Fig. 1A). Second, the mice performed each of four basic motor behaviors (standing on a horizontal surface, postural corrections on a tilting platform, forward locomotion, and righting) before and during unilateral activation/inactivation of CaMKIIa-RNs caused by CNO injection. Motor performance before and during activation/inactivation of CaMKIIa-RNs were analyzed and compared. In the following text, terms "ipsilateral" and "contralateral" are used to indicate, respectively, the ipsilateral and contralateral side in relation to the side of the virus injection.

The same viral titer, injection volume, and injection coordinates were used for all animals (see "Methods" for detail). A representative

https://doi.org/10.1038/s42003-025-08967-z                                                    **Article**

example of an infected area (CaMKIIa-RNs expressing DREADDs) is shown in Fig. 1B. The infected area covered mainly the medullary reticular nucleus ventral part (MdV) and the intermediate reticular nucleus (IRt), as seen in the heatmap of the intensity of the mCherry fluorescence averaged across all animals (Fig. 1C). By targeting the MdV-IRt area, we ensured that we manipulate a specific subpopulation of CaMKIIa-RNs within the overall reticular formation, thereby allowing us to investigate their distinct functional roles in postural control. The specificity of DREADDs expression in CaMKIIa-RNs was confirmed by co-localization of CaMKIIa mRNA and mCherry (cells indicated by white arrowheads in Fig. 1D). The representative example of rostro-caudal extent of a viral infection area is shown in Fig. 1E. It largely covered the caudal half of MdV rostro-caudal extend. The infected area was similar across animals.

### Left-right asymmetry in activity of CaMKIIa-RNs evokes the body roll tilt in the animal standing on a horizontal surface

To reveal changes in the basic body posture of the mouse standing on a horizontal surface, the frontal and rear views of the mouse (Fig. 2A, D) as well as its view from below (Fig. 2G) were video recorded before and during unilateral activation/inactivation of CaMKIIa-RNs (respectively, Control and *CNO* in Fig. 2A, D, G). We found that, while before CNO administration animals maintained the dorsal side-up orientation of the head and trunk (Control in Fig. 2A, D), injection of CNO caused a gradually developing roll tilt of the head and trunk. The asymmetry began to emerge in 15–20 min post-injection, reached its maximal expression in approximately 40 min after CNO injection and persisted for about 1–1.5 h (*CNO* in Fig. 2A, D). During this period, animals with more pronounced asymmetry, from time to time actively attempted to reduce it, but were unable to correct fully the asymmetry. Recovery was gradual, with a progressive return to symmetrical posture in 3–4 h after the injection.

We found that unilateral activation of CaMKIIa-RNs evoked ipsilateral (in relation to the side of the activated neurons) roll tilt of the head and trunk (compare Control and CNO in Fig. 2A, Supplementary Video 1). On average, a significant increase in the values of the ipsilateral head and trunk tilt angles during unilateral activation of CaMKIIa-RNs, as compared to those in control were observed ($-30 \pm 17°$ vs $-4 \pm 6°$ for the head tilt angle and $-31 \pm 10°$ vs $1 \pm 5°$ for the trunk tilt angle; paired $t$ test, $p = 9 \times 10^{-4}$ and $p = 6 \times 10^{-5}$, respectively; Fig. 2B).

The ipsilateral roll tilt of the trunk was caused by a change in configuration of the left and right limbs as well as in their position in relation to the trunk. To reveal asymmetry in the configuration of the ipsilateral and contralateral limbs caused by unilateral activation of CaMKIIa-RNs, we compared the limbs length as well as their lateral positions in relation to the trunk before and after CNO injection.

To estimate the asymmetry in the limbs length, we calculated the extension/flexion asymmetry index for hindlimbs during standing based on the difference between lengths of the contralateral and ipsilateral limb (see Methods for details). Positive values of the extension/flexion asymmetry index indicate greater extension of contralateral limb and negative values reflect greater extension of ipsilateral limb. We found that in control, the asymmetry index was close to 0 ($-0.01 \pm 0.03$), indicating that the lengths of the left and right limbs were almost equal (Fig. 2C). By contrast, during unilateral activation of CaMKIIa-RNs, the asymmetry index was positive ($0.24 \pm 0.08$, statistically different from Control, paired $t$ test, $p = 3 \times 10^{-4}$), indicating that the contralateral limb was longer (more extended) than the ipsilateral one (Fig. 2C). Thus, unilateral activation of CaMKIIa-RNs evoked asymmetry in configuration of the ipsilateral and contralateral hindlimbs – the contralateral limb became more extended than the ipsilateral one.

To estimate the asymmetry in the lateral positions of the limbs in relation to the trunk, we calculated the abduction/adduction asymmetry index for forelimbs and hindlimbs (see "Methods" for details). Positive values of the abduction/adduction asymmetry index indicate greater

abduction of the contralateral limb, and negative values reflect greater abduction of the ipsilateral limb. In control, the lateral positions of the left and right limbs in relation to the trunk midline were almost symmetrical (Fig. 2G, left panel) and the values of the abduction/adduction asymmetry index for both forelimbs and hindlimbs were close to 0 (Fig. 2J, K). By contrast, during unilateral activation of CaMKIIa-RNs, the lateral positions of the left and right limbs were highly asymmetrical: both the ipsilateral forelimb and hindlimb were closer to the trunk midline than the contralateral limbs (Fig. 2G, right panel). On average, the lateral positions of the contralateral forelimb and hindlimb (expressed in percent of the half of the corresponding body width) were significantly larger than in control (respectively, $144 \pm 30\%$ vs $70 \pm 9\%$ and $206 \pm 39\%$ vs $110 \pm 21\%$; paired $t$ test, $p = 6 \times 10^{-4}$ and $p = 0.001$, Fig. 2H). By contrast, the lateral positions of the ipsilateral forelimb as well as hindlimb were on average smaller than in control and significant only for forelimb (respectively, $-16 \pm 40\%$ and $-64 \pm 17\%$ vs $-89 \pm 40\%$ and $-106 \pm 18\%$; paired $t$ test, $p = 0.03$ and $p = 0.34$; Fig. 2H). The asymmetry in the lateral positions of the left and right limbs was also reflected in the values of the abduction/adduction asymmetry index that were significantly higher than those in control for both fore- and hindlimbs (respectively, $0.92 \pm 0.47$ vs $0.07 \pm 0.11$ and $0.46 \pm 0.20$ vs $0.01 \pm 0.08$, paired $t$ test, $p = 0.002$ and $p = 7 \times 10^{-4}$; Fig. 2J). Note that after CNO the values of the abduction/adduction asymmetry indexes were positive indicating that contralateral limbs were more abducted than ipsilateral ones. Thus, these results suggest that unilateral activation of CaMKIIa-RNs evokes left-right asymmetry in position of limbs in relation to the trunk caused by abduction of the contralateral limbs and adduction of the ipsilateral forelimbs.

Unilateral inactivation of CaMKIIa-RNs evoked effects opposite to those observed during unilateral CaMKIIa-RNs activation. It resulted in the contralateral (in relation to the inactivated neurons) roll tilt of the head and trunk (Fig. 2D, E, Supplementary Video 2) caused by extension and abduction of ipsilateral limbs as well as by flexion and adduction of the contralateral limbs (Fig. 2F, I). On average, parameters such as head and trunk roll angles and the limbs extension/flexion asymmetry index were significantly different compared to those in control (head angle: $10 \pm 9°$ after CNO vs $-1 \pm 4°$ in control; trunk angle: $12 \pm 7°$ after CNO vs $-1 \pm 3°$; extension/flexion asymmetry index: $-0.06 \pm 0.06$ after CNO vs $0.01 \pm 0.02$ in control; paired $t$ test, $p = 9 \times 10^{-4}$, $p = 6 \times 10^{-5}$, $p = 0.02$, respectively; Fig. 2E, F). However, the absolute values of their changes were almost four times smaller than during unilateral activation of CaMKIIa-RNs (compare the corresponding parameters in **E** and **B, F** and **C** in Fig. 2). Also, asymmetry in the positions of the left and right limbs in relation to the trunk were much weaker expressed during unilateral CaMKIIa-RNs inactivation (Fig. 2I) as compared to those observed during unilateral activation (Fig. 2H). In particular, the changes in abduction of both forelimb and hindlimb as compared to control were significantly larger during unilateral activation than during unilateral inactivation of CaMKIIa-RNs ($0.37 \pm 0.15$ vs $0.14 \pm 0.10$ for forelimbs and $0.48 \pm 0.22$ vs $0.23 \pm 0.15$ for hindlimbs, unpaired $t$ test, $p = 0.0087$ and $p = 0.034$, respectively), while corresponding changes in adduction were non-significant ($0.29 \pm 0.13$ vs $0.25 \pm 0.11$ for forelimbs and $0.18 \pm 0.12$ vs $0.08 \pm 0.04$ for hindlimbs, unpaired $t$ test, $p = 0.56$ and $p = 0.08$, respectively). Nevertheless, values of abduction of the ipsilateral limbs and adduction of the contralateral limbs during unilateral CaMKIIa-RNs inactivation were significant as compared to control for the hindlimbs (respectively, $-153 \pm 17\%$ vs $-110 \pm 18\%$ and $99 \pm 21\%$ vs $114 \pm 14\%$; paired $t$ test, $p = 0.03$ and $p = 0.03$, Fig. 2I). Despite the weaker effect, inactivation of CaMKIIa-RNs evoked a significant asymmetry in the lateral position of the left and right limbs. During inactivation, the values of the abduction/adduction asymmetry index for both fore- and hindlimbs differed significantly from those in control (respectively, $-0.79 \pm 0.63$ vs $0.03 \pm 0.15$ and $-0.24 \pm 0.11$ vs $0.01 \pm 0.04$, paired $t$ test, $p = 0.003$ and $p = 0.003$; Fig. 2K) and had negative values indicating that the ipsilateral limbs were more abducted than contralateral ones.

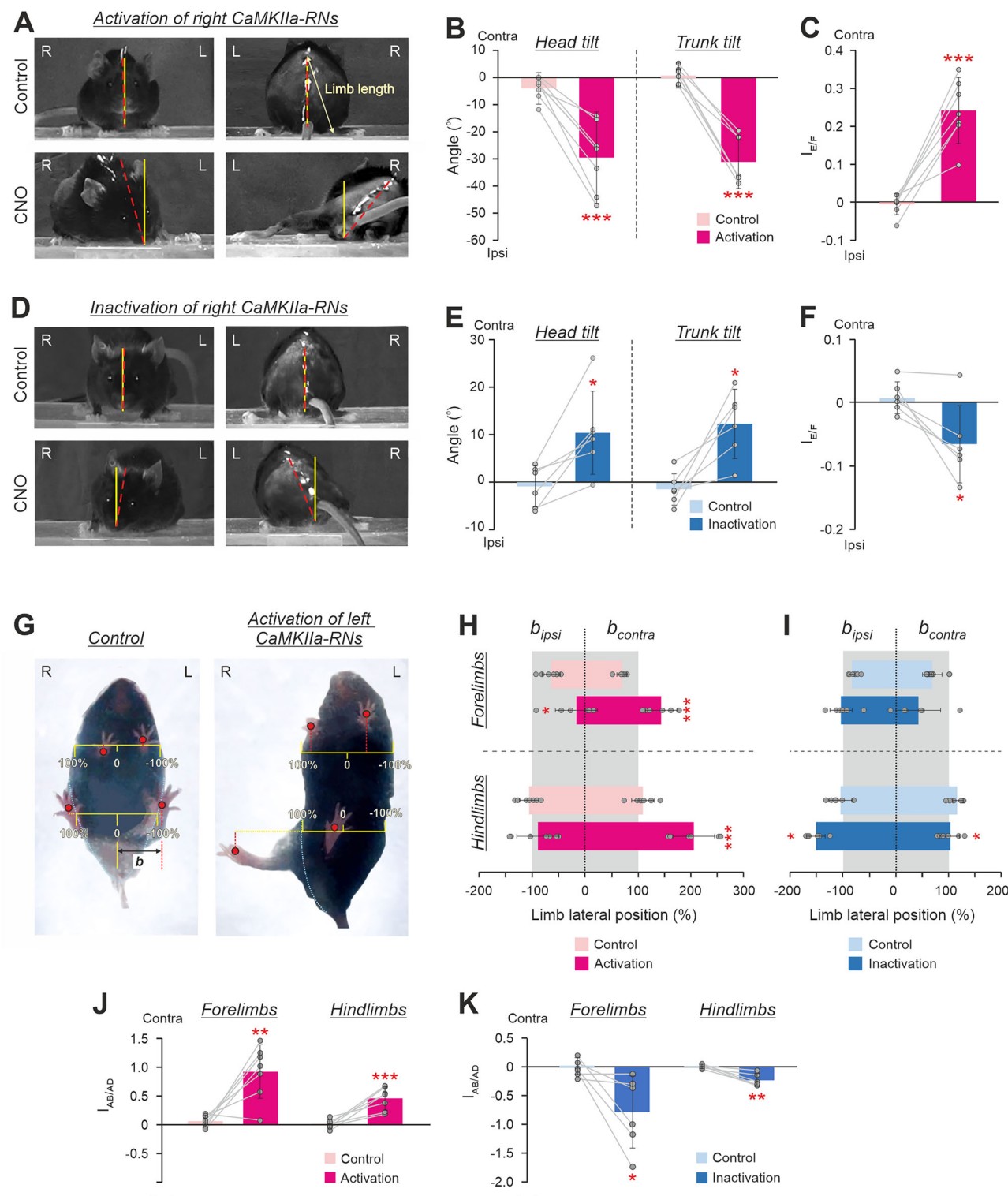

**Fig. 2 | Effects of unilateral activation/inactivation of CaMKIIa-RNs on the body posture in mice standing on a horizontal surface. A**, **D** The front and rear views of mice before (Control) and during activation (**A**, *CNO*) and inactivation (**D**, *CNO*) of the right CaMKIIa-RNs. The solid yellow line indicates the vertical. The dashed red line indicates the dorso-ventral axis of the head or trunk. **B**, **E** Values of the head and the trunk roll tilt angles in individual animals, as well as corresponding mean ± SD values, before (Control) and during unilateral activation (**B**) and inactivation (**E**) of CaMKIIa-RNs. **C**, **F** Values of the extension/flexion asymmetry index ($I_{E/F}$) in individual animals, as well as the corresponding mean ± SD values, before (Control) and during unilateral activation (**C**) and inactivation (**F**) of CaMKIIa-RNs. **G** The view from below of a mouse before (Control) and during activation of the left CaMKIIa-

RNs. **H**, **I** The lateral positions of the contralateral and the ipsilateral limbs in individual animals as well as the corresponding mean ± SD values, before (Control) and during unilateral activation (**H**) and inactivation (**I**) of CaMKIIa-RNs. The lateral position is measured in percents of the corresponding half body width. **J**, **K** The abduction/adduction asymmetry index of the forelimbs and hindlimbs ($I_{AB/AD}$) in individual animals, as well as the corresponding mean ± SD values, before (Control) and during unilateral activation and inactivation of CaMKIIa-RNs. CaMKIIa-RNs activation: **B**, **C**, **H**, **J**, $N = 7$. CaMKIIa-RNs inactivation: **E**, **F**, **I**, **K**, $N = 6$. L and R, left and right, respectively. *FL* and *HL*, forelimb and hindlimb, respectively. *Ipsi* and *Contra*, ipsilateral and contralateral in relation to the virus injection side, respectively. Indication of significance level: * $0.01 < p < 0.05$, ** $0.001 < p < 0.01$, *** $p < 0.001$.

**Article**

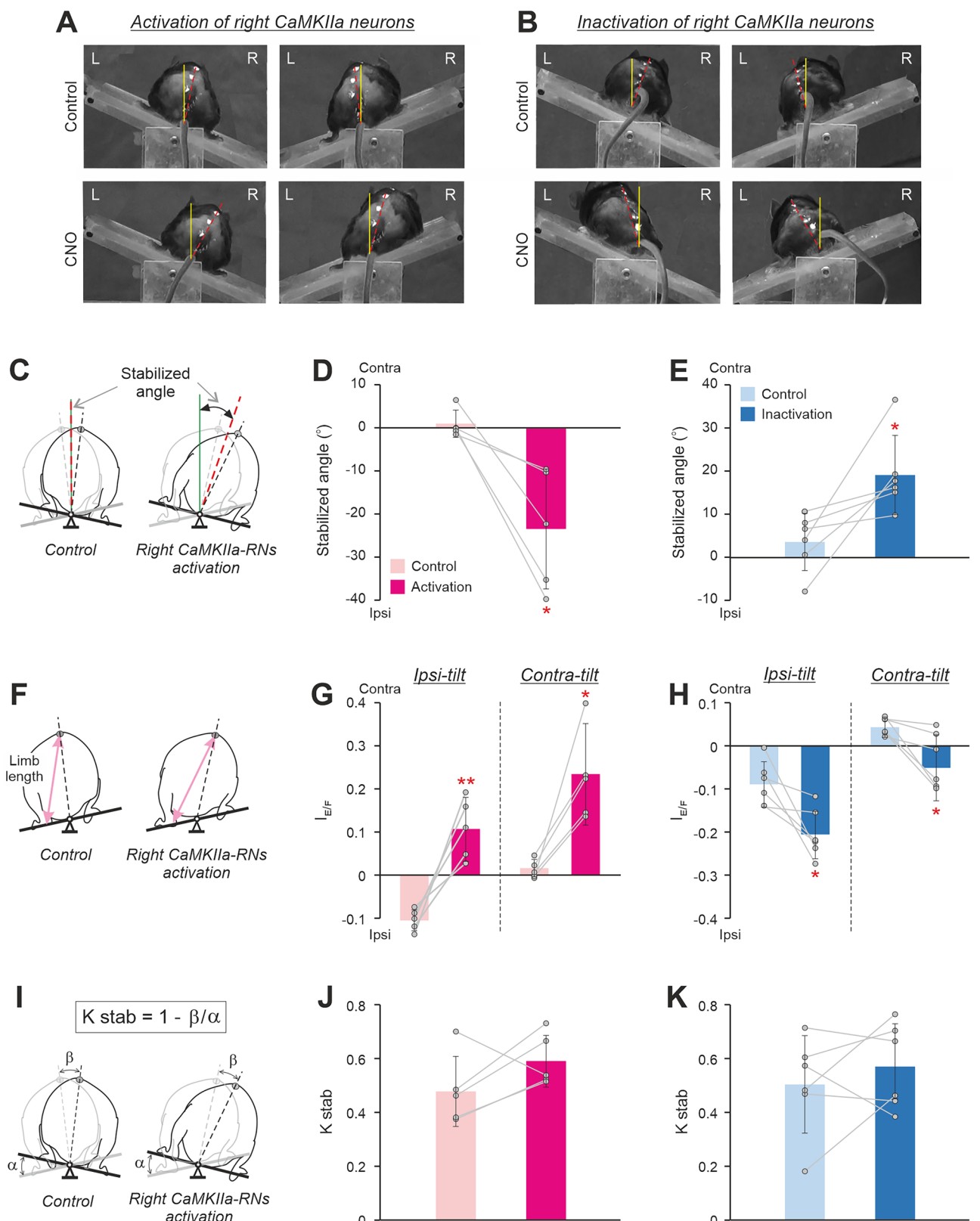

In summary, a left-right asymmetry in activity of CaMKIIa-RNs located in MdV-IRt area evoked the body roll tilt toward more active (dominant) sub-population of CaMKIIa-RNs. The body roll tilt was caused by flexion and adduction of limbs on the dominant side and by simultaneous extension and abduction of the limbs on the opposite side. The effects of activation of the CaMKIIa-RNs were stronger than those of inactivation.

**The roll tilt caused by asymmetry in activity of CaMKIIa-RNs is actively stabilized during standing on the tilting platform**

To find out whether the body roll tilt evoked by unilateral activation/inactivation of CaMKIIa-RNs during standing on a horizontal surface is actively stabilized, we analyzed postural corrections of mice standing on the platform subjected to lateral tilts (Fig. 3).

**Fig. 3 | Effects of unilateral activation/inactivation of CaMKIIa-RNs on postural corrections caused by lateral tilts of the platform. A**, **B** The rear views of mice standing on the platform tilted to the right (left panels) and to the left (right panels) before (Control) and after activation (**A**, *CNO*) and inactivation (**B**, *CNO*) of the right CaMKIIa-RNs. Designations as in Fig. 2A, B. **C** Schematic drawings indicating the "*Stabilized angle*" on the tilting platform before (*Control*) and during activation of the right CaMKIIa-RNs (see "Methods" for details). The black and gray dashed lines indicate the dorso-ventral axis of the trunk when the animal is standing on the platform tilted to the right and to the left, respectively. The green solid line indicates the vertical. The dashed crimson line indicates the bisector of the angle formed by the dorso-ventral axis of the trunk at two conditions: when the animal is standing on the platform tilted to the left and when it standing on the platform tilted to the right. **D**, **E** Values of the stabilized angle in individual animals, as well as the corresponding mean ± SD values, before (Control) and during unilateral activation (**D**) and inactivation (**E**) of CaMKIIa-RNs. **F** Schematic drawings indicating left hindlimb lengths (shown by the pink line with arrows) during standing on the platform tilted to the left before (Control) and during activation of the right CaMKIIa-RNs. **G**, **H** Values of the extension/flexion asymmetry index during standing ($I_{E/F}$) in individual animals, as well as the corresponding mean ± SD values, during standing, respectively, on the ipsilaterally and contralaterally tilted platform (Ipsi-tilt and Contra-tilt, respectively) before (Control) and after unilateral activation (**G**) and inactivation (**H**) of the CaMKIIa-RNs. **I** Schematic drawings explaining the estimation of the efficacy of postural corrections ($K_{STAB}$). α, the amplitude of the platform tilt; β, the amplitude of the dorso-ventral axis of the trunk tilt during standing on the tilting platform. **J**, **K** Values of the Kstab in individual animals, as well as the corresponding mean ± SD values, before (Control) and during unilateral activation (**J**) and inactivation (**K**) of CaMKIIa-RNs. In **D**, **G**, **J**: *N* = 5. In **E**, **H**, **K**: *N* = 6. Abbreviations as in Fig. 2.

Before CNO injection, a lateral tilt of the supporting platform evoked extension of limbs on the side of the tilt and simultaneous flexion of the contralateral limbs (Control in Fig. 3A, B, Supplementary Video 3) leading to a displacement of the dorso-ventral axis of the trunk (indicated by the red dashed line in Fig. 3A, B) towards the vertical (indicated by the solid yellow line in Fig. 3A, B). However, as in all tested terrestrial quadrupeds[17,34,35], the postural corrections in mice did not fully compensate the distortion of the trunk orientation caused by the platform tilt, and after their execution, the dorso-ventral axis of the trunk was still deviated from the vertical (Control in Fig. 3A, B)[36].

The orientation of the trunk stabilized on the tilting platform was characterized by a "Stabilized angle" defined as the average orientation between tilts to the left and right. Negative values indicate stabilization with an ipsilateral roll tilt, while positive values reflect stabilization with a contralateral roll tilt (see Methods for details; Fig. 3C). Before CNO injection, the stabilized angle was close to 0 suggesting that the animal stabilized close to the dorsal-side-up trunk orientation (Control in Fig. 3D, E, Supplementary Video 3). We found that during unilateral activation of CaMKIIa-RNs, the value of the stabilized angle was negative and significantly different from the control (-24 ± 14° *vs* 1 ± 3°, paired *t* test, *p* = 0.02; Fig. 3D, Supplementary Video 3) suggesting that the animal stabilized the body orientation with an ipsilateral roll tilt. By contrast, during unilateral inactivation, the stabilized angle value was positive and significantly different from that in control (19 ± 9° *vs* 4 ± 7°, paired *t* test, *p* = 0.03; Fig. 3E) suggesting that the animal stabilized the body orientation with a contralateral roll tilt.

The change of the stabilized trunk orientation was caused by changes in configurations of the left and right limbs performing the corrective movements. In control, at condition when the contralateral and ipsilateral limb were standing on the side of the tilt, the averaged extension/flexion asymmetry index had, respectively, positive and negative values, indicating that the length of the limb on the side of the tilt was longer than the length of the opposite limb (Control in Fig. 3G, H). The increase in the limb length (the limb extension) on the side of the tilt and the simultaneous decrease in the length (flexion) of the opposite limb moved the dorso-ventral axis of the trunk toward the vertical. We found that during unilateral activation of CaMKIIa-RNs, the mean value of the extension/flexion asymmetry index was positive during both ipsilateral and contralateral tilts (Fig. 3G). Also, the values of the extension/flexion asymmetry index differed significantly from those in control (0.11 ± 0.07 vs −0.10 ± 0.02 for the ipsilateral tilt and 0.24 ± 0.12 vs 0.01 ± 0.02 for the contralateral tilt; paired *t* test, *p* = 0.005 and *p* = 0.02, respectively; Fig. 3G).

These results suggest that asymmetry in the hindlimbs length evoked by unilateral activation of CaMKIIa-RNs in animals standing on a horizontal surface was maintained during postural corrections. During both ipsilateral and contralateral tilt, the length of the contralateral limb was larger than that of the ipsilateral one. Thus, contralateral and ipsilateral limb performed corrective movements with more extended and flexed configuration, respectively, as compared to control. This led to the positive value of the stabilized angle, reflecting stabilization of the trunk orientation with the ipsilateral roll tilt.

We found that unilateral inactivation of CaMKIIa-RNs led to the opposite effects (Fig. 3H, Supplementary Video 4). During both ipsilateral and contralateral tilts, the mean values of the extension/flexion asymmetry index were negative and significantly different from corresponding values in control (ipsilateral tilt: −0.20 ± 0.06 during CNO vs −0.09 ± 0.05 in control, paired *t* test, *p* = 0.03; contralateral tilt: −0.05 ± 0.08 during CNO vs 0.04 ± 0.02 in control, paired *t* test, *p* = 0.03). Thus, the ipsilateral limb performed corrective movements with more extended configuration, while the contralateral limb performed postural corrections with more flexed configuration as compared to those in control. This led to the negative value of the stabilized angle, reflecting stabilization of the trunk orientation with the contralateral roll tilt (Fig. 3E).

To reveal possible effects of unilateral activation/inactivation of CaMKIIa-RNs on the efficacy of postural corrections stabilizing the trunk orientation, we calculated the coefficient of postural stabilization (Fig. 3I). We found that the mean value of the coefficient of stabilization during unilateral activation (Fig. 3J) as well as during unilateral inactivation (Fig. 3K) of CaMKIIa-RNs did not differ significantly from that in control. These results suggest that asymmetry in activity of CaMKIIa-RNs does not affect the efficacy of postural corrections.

## The roll tilt caused by asymmetry in activity of CaMKIIa-RNs is maintained during locomotion

Next, we addressed the question, whether the roll tilt caused by unilateral activation/inactivation of CaMKIIa-RNs is maintained during locomotion. As in other terrestrial quadrupeds[9,37,38], in mice, there are left-right oscillations of the spine during locomotion with the maximal deviation of the spine toward the hindlimb that is in the beginning of the stance phase of locomotor cycle while the opposite hindlimb is at the end of the stance (at toe-off moment; Fig. 4A). To find out whether there is lateral displacement of these spine oscillations during activation/inactivation of CaMKIIa-RNs as compared to control, we calculated the stabilized spine position before and after CNO administration (see Methods for details). A displacement of the spine position towards the lateral edge of the body outline on the top view indicates that the body orientation was maintained with the roll tilt. We found that during unilateral activation of CaMKIIa-RNs, the spine displacement values were negative for most animals indicating that the spine was displaced toward the ipsilateral side and thus, ipsilateral body roll tilt was maintained. However, on average, the displacement was not significantly different from control (−6.1 ± 7.1% vs −0.5 ± 3.1%, paired *t* test, *p* = 0.16; Fig. 4D). By contrast, during unilateral inactivation of CaMKIIa-RNs, the values of the spine displacement were positive, and the population average of the displacement was significantly different from control (−3.3 ± 2.8% vs 2.2 ± 2.2%, paired *t* test, *p* = 0.0013; Fig. 4D). Thus, unilateral inactivation of CaMKIIa-RNs during locomotion evoked displacement of the spine toward the contralateral side suggesting that the trunk orientation with a contralateral roll tilt was maintained.

To examine whether the asymmetry in the lateral position of the left and right limbs caused by the unilateral activation/inactivation of CaMKIIa-RNs in standing animals was maintained during locomotion, we calculated

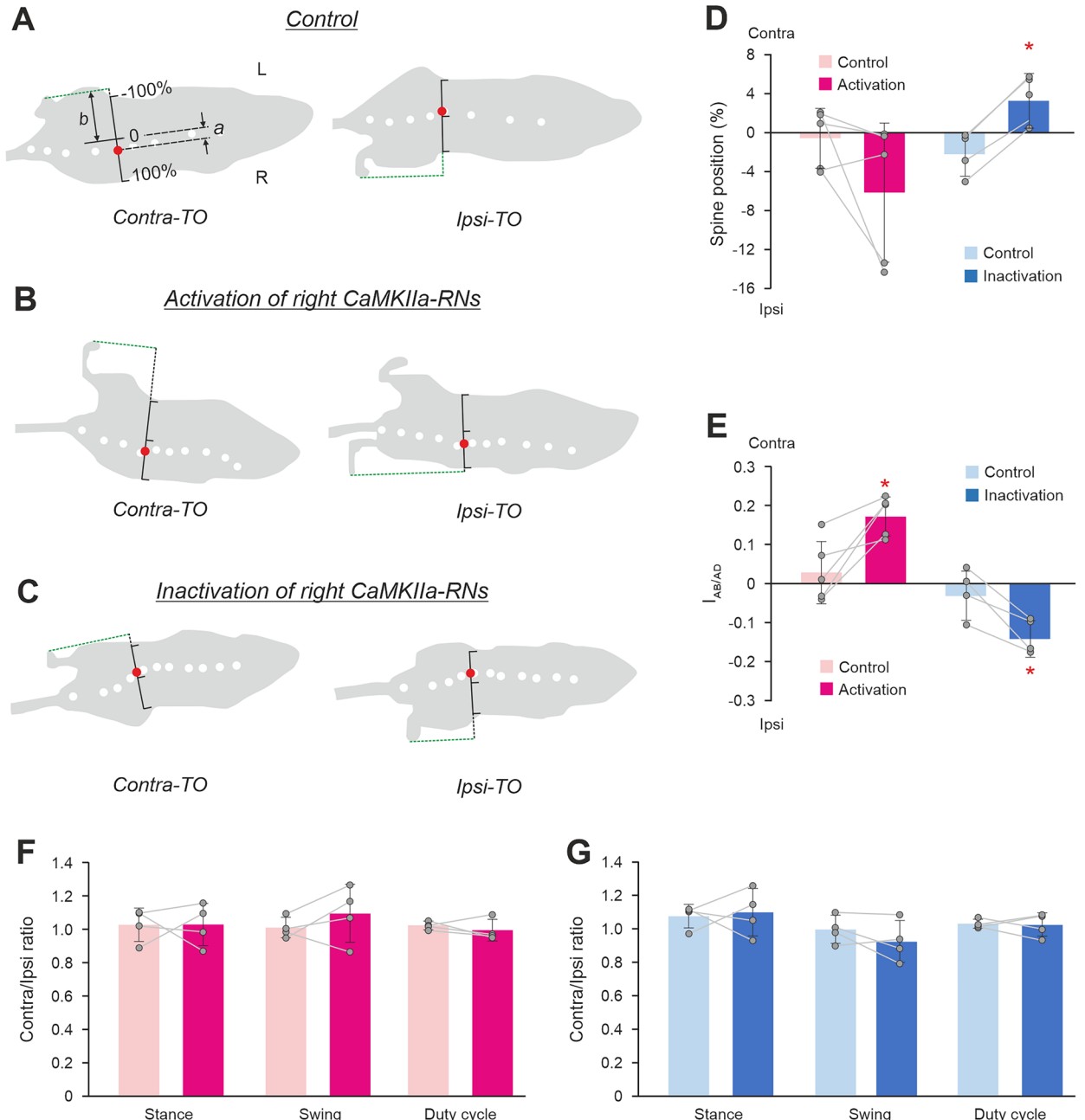

**Fig. 4 | Effects of unilateral activation/inactivation of CaMKIIa-RNs on the body orientation during locomotion.** A–C Silhouettes of the top views of the walking mouse at the moments of the maximal displacement of the spine toward the ipsilateral hindlimb [at the moment of the contralateral hindlimb toe off (Contra-TO), left panels] and toward the contralateral limb [at the moment of the ipsilateral hindlimb toe off (*Ipsi-TO*), right panels] in control (**A**) and during activation (**B**) and inactivation (**C**) of the right CaMKIIa-RNs. White dots are markers on the spine. The red dot indicates the point on the spine that exhibits maximal left-right oscillations during locomotion. The black scale indicates the body width with its middle considered as "0" and the ipsilateral and contralateral edges of the body as +100% and −100%, respectively. **D**, **E** Values of the spine position (a displacement of the spine from the left-right body edges midline; **D**) and abduction/adduction asymmetry index ($I_{AB/AD}$; **E**) during locomotion in individual animals, as well as the corresponding mean ± SD values, before (Control) and during unilateral activation and inactivation of CaMKIIa-RNs. **F,G** Values of the ratio between locomotor parameters (stance durations, swing durations, and duty cycles) of the contralateral and ipsilateral hindlimbs in individual animals, as well as the corresponding mean ± SD values, before (Control) and during unilateral activation and inactivation of CaMKIIa-RNs. In **D**, **E**: $N = 5$, $n = 40$ for control and $n = 43$ for unilateral activation; $N = 4$, $n = 47$ for control and $n = 41$ for unilateral inactivation. In **F**, **G**: $N = 4$, $n = 39$ for unilateral activation and $N = 4$, $n = 42$ for unilateral inactivation. In **D**, **E** Abbreviations as in Fig. 2.

the abduction/adduction asymmetry index during locomotion in control and after activation/inactivation of CaMKIIa-RNs (see Methods for details). Positive and negative values of the index indicate displacement of the limbs from their symmetrical position in relation to the trunk toward the contralateral and ipsilateral side, respectively. We found that after unilateral activation of CaMKIIa-RNs, the values of the abduction/adduction asymmetry index were positive and the mean value of the index was significantly higher than that in control (respectively, $0.17 \pm 0.05$ *vs* $0.03 \pm 0.08$; paired *t* test, $p = 0.02$; Fig. 4E). These results suggest that unilateral activation of CaMKIIa-RNs evoked displacement of the hindlimbs toward the contralateral side in contrast to almost symmetrical limb position observed in control. On the other hand, after unilateral inactivation of CaMKIIA-RNs, values of the abduction/adduction asymmetry index were negative (Fig. 4E). On average, the mean value of the index differed significantly from that in

control (respectively, −0.14 ± 0.05 vs −0.03 ± 0.06; paired *t* test, *p* = 0.02; Fig. 4E). These results suggest that unilateral inactivation of CaMKIIa-RNs evoked displacement of the hindlimbs toward the ipsilateral side as compared to control. Thus, during locomotion, lateral displacement of the limbs in relation to the trunk toward the contralateral side during CaMKIIa-RNs activation may contribute to maintenance of the trunk orientation with some roll tilt to the ipsilateral side, while displacement of limbs toward the ipsilateral side during CaMKIIa-RNs inactivation may contribute to maintenance of the contralateral tilt of the trunk.

To assess whether asymmetry in activity of CaMKIIa-RNs induced asymmetry in temporal parameters of locomotor movements of the left and right limbs, we calculated the ratio between values of the stance duration, swing duration, and duty cycle of the contralateral and ipsilateral hindlimbs (see Methods for details). We found that after unilateral activation or inactivation of CaMKIIa-RNs, all these ratios were close to 1 and did not significantly differ from corresponding control values (paired *t* test, all *p* > 0.05; Fig. 4F, G). These results indicate that asymmetry in activity of CaMKIIa-RNs did not evoke asymmetry in temporal parameters of locomotor movements of the left and right limbs.

However, we found that both unilateral activation and inactivation of CaMKIIa-RNs resulted in an increase of the cycle duration due to an increase in the duration of stance, and thus a decrease of speed, while the duration of swing remained similar to that in control. During inactivation, the changes in cycle and stance durations, as well as speed, were significant (respectively, 0.54 ± 0.03 s *vs* 0.38 ± 0.04 s, 0.40 ± 0.02 s vs 0.26 ± 0.04 s, and 8.25 ± 3.04 cm/s *vs* 11.47 ± 2.31 cm/s; paired *t* test, *p* = 0.02, *p* = 0.01, and *p* = 0.0054), and non-significant for swing duration (0.14 ± 0.03 s *vs* 0.12 ± 0.01 s; paired *t* test, *p* = 0.26). During activation, the changes were non-significant (for cycle, stance, swing durations, and speed respectively, 0.57 ± 0.09 s vs 0.38 ± 0.09 s, 0.45 ± 0.09 s vs 0.26 ± 0.07 s, 0.12 ± 0.01 s vs 0.12 ± 0.02 s, and 6.84 ± 2.02 cm/s *vs* 10.02 ± 3.14 cm/s; paired *t* test, *p* = 0.07, *p* = 0.06, *p* = 0.84, and *p* = 0.06).

## Asymmetry in activity of CaMKIIa-RNs hinders execution of righting behavior

The righting behavior[39] that requires coordinated activity of left and right muscles of the trunk and limbs, is essential for control of posture. To find out whether asymmetry in CaMKIIa-RNs affects its execution, we compared the righting behavior before and during unilateral CaMKIIa-RNs activation/inactivation.

To evoke the righting behavior, we released animals in an upside-down position (moment 1 in Fig. 5A). In control, mice performed the righting in two Stages[36]. During Stage 1, twisting and lateral bending (oblique bending) of the forequarters in relation to the hindquarters led to rotation of the body toward the side of twisting that was accompanied by movements of the forelimbs toward the surface. At the end of the Stage 1, the forequarters assumed a position with forelimbs standing on the surface, while the hindquarters turned from the upside-down position to the side (moment 2 in Fig. 5A). During Stage 2, the hindquarters rotated in relation to the forequarters until they reached a position close to the dorsal side up with hindlimbs standing on the surface (moment 3 in Fig. 5A).

Out of 13 tested mice, 11 mice were able to successfully perform both Stages of the righting with rotation to the ipsilateral (upward movement of the infected side) as well as to contralateral (downward movement of the infected side) side during both unilateral activation (N = 6) and unilateral inactivation (*N* = 5) of CaMKIIa-RNs. Two mice with excitatory DREADDs, during unilateral activation of CaMKIIa-RNs, performed ipsilateral righting but were unable to perform the righting to the contralateral side. We defined the righting duration as the sum of Stage 1 and Stage 2 durations.

We found that during unilateral activation of CaMKIIa-RNs, righting to the contralateral side was performed significantly slower than in control (0.25 ± 0.06 vs 0.18 ± 0.01; paired *t* test, *p* = 0.04; Fig. 5B). However, the duration of the ipsilateral righting was similar to that in control (0.17 ± 0.02 s vs 0.17 ± 0.04 s; paired *t* test, *p* = 0.95; Fig. 5B).

During unilateral inactivation of CaMKIIa-RNs, righting to both the ipsilateral and contralateral side was performed slower than in control (ipsilateral: 0.22 ± 0.04 s vs 0.17 ± 0.02 s; contralateral: 0.21 ± 0.03 s vs 0.19 ± 0.04 s; Fig. 5C). However, these changes did not reach statistical significance (paired *t* test, *p* = 0.16 and *p* = 0.36 for ipsilateral and contralateral righting, respectively).

Thus, there was a tendency that left-right asymmetry in activity of CaMKIIa-RNs led to an increase in the duration of the righting performed toward the less active subpopulation of CaMKIIa-RNs. To clarify whether this increase in the duration was caused by asymmetry in the left-right limb configurations, we plotted the abduction/adduction asymmetry index as well as the extension/flexion asymmetry index against the change in the righting duration performed toward the side with lower CaMKIIa-RN activity (Fig. 5D, E, respectively). We found a significant positive correlation between parameters in both cases suggesting that asymmetry in configurations of the left and right limbs caused by unilateral activation/inactivation of CaMKIIa-RNs – extension and abduction of the hindlimb on the side of the less active subpopulation of CaMKIIa-RNs and simultaneous flexion and adduction of the hindlimb on the opposite side – distorted righting reflex toward the side with lower CaMKIIa-RN activity.

## The body roll tilt was caused specifically by CaMKIIa-RNs located in MdV-IRt area of the caudal medulla

To clarify whether the effects of CaMKIIa-RNs on the body orientation in the transverse plane was specific for MdV–IRt area of the caudal medulla, we studied effects of unilateral activation of CaMKIIa-RNs in the gigantocellular reticular nuclei, that is located rostrally to MdV-IRt (Gi, Fig. 6A) and in the area that is dorsal to MdV-IRt (d-MdV, Fig. 6D). The same viral titer and injection volume as during injection to MdV-IRt area were used for all animals (see Methods for detail). Figure 6B,C,E,F shows representative examples of areas infected in Gi (**B**) and in d-MdV (**E**), as well as the average infected areas in Gi (**C**) and in d-MdV (**F**) presented as heatmaps.

We found that unilateral activation of CaMKIIa-RNs in these two areas produced no substantial effects on the body orientation in the transverse plane. During standing on a horizontal surface, for the roll tilt angles of the head and trunk, difference from the corresponding control values was on average close to 0 (95% confidence intervals were from −0.7° to 3.5° for the head tilt and from −3.8° to 0.4° for the trunk tilt during unilateral activation of CaMKIIa-RNs in Gi, from −0.1° to 3.6° for the head tilt and from −2.9° to 4.1° for the trunk tilt during unilateral activation of CaMKIIa-RNs in d-MdV; Fig. 6G, H). By contrast, the differences during unilateral activation of CaMKIIa-RNs in MdV–IRt area and control were large and statistically significant both for the head and trunk tilts (respectively, −25.6 ± 11° and −31.8 ± 8.49°, one-sample *t* test, *p* = 8 × 10⁻⁴, and *p* = 6 × 10⁻⁴; Fig. 6G, H). These results suggest that CaMKIIa-RNs in MdV–IRt area but not in the adjacent areas, contribute to control of body orientation in the transverse plane.

Next, we addressed the question whether the effects on the body orientation in the transverse plane are specific to the molecular identity of CaMKIIa-RNs in MdV-IRt area. To answer this question, first, we compared the effects of unilateral activation of a broad population of glutamatergic (Vglut2) neurons and CaMKIIa-RNs in MdV-IRt area. For this purpose, we infected Vglut2 reticular neurons (Vglut2-RNs) with AVV-hsyn-DIO-hM3Dq-mCherry in MdV-IRt area of Vglut2-Cre mice (Fig. 7A, B). Figure 7D–H compares effects of unilateral activation of Vglut2-RNs and CaMKIIa-RNs in MdV-IRt area. While activation of CaMKIIa-RNs evoked the ipsilateral roll tilt of the body (reflected in displacement of the spine toward the ipsilateral side of the body; right panels in Fig. 7G, H), activation of Vglut2-RNs caused the ipsilateral bending of the body in the yaw plane (left panels in Fig. 7G, H). Also, during unilateral activation of CaMKIIa-RNs, the animal was able to stand still without the head or limb movements (Fig. 7D, E, lower panels). By contrast, during unilateral activation of Vglut2-RNs, continuous movements of the head and ipsilateral forelimb (Fig. 7D, E, upper panels) accompanied standing. Finally, during unilateral activation of Vglut2-RNs, the animals performed continuous

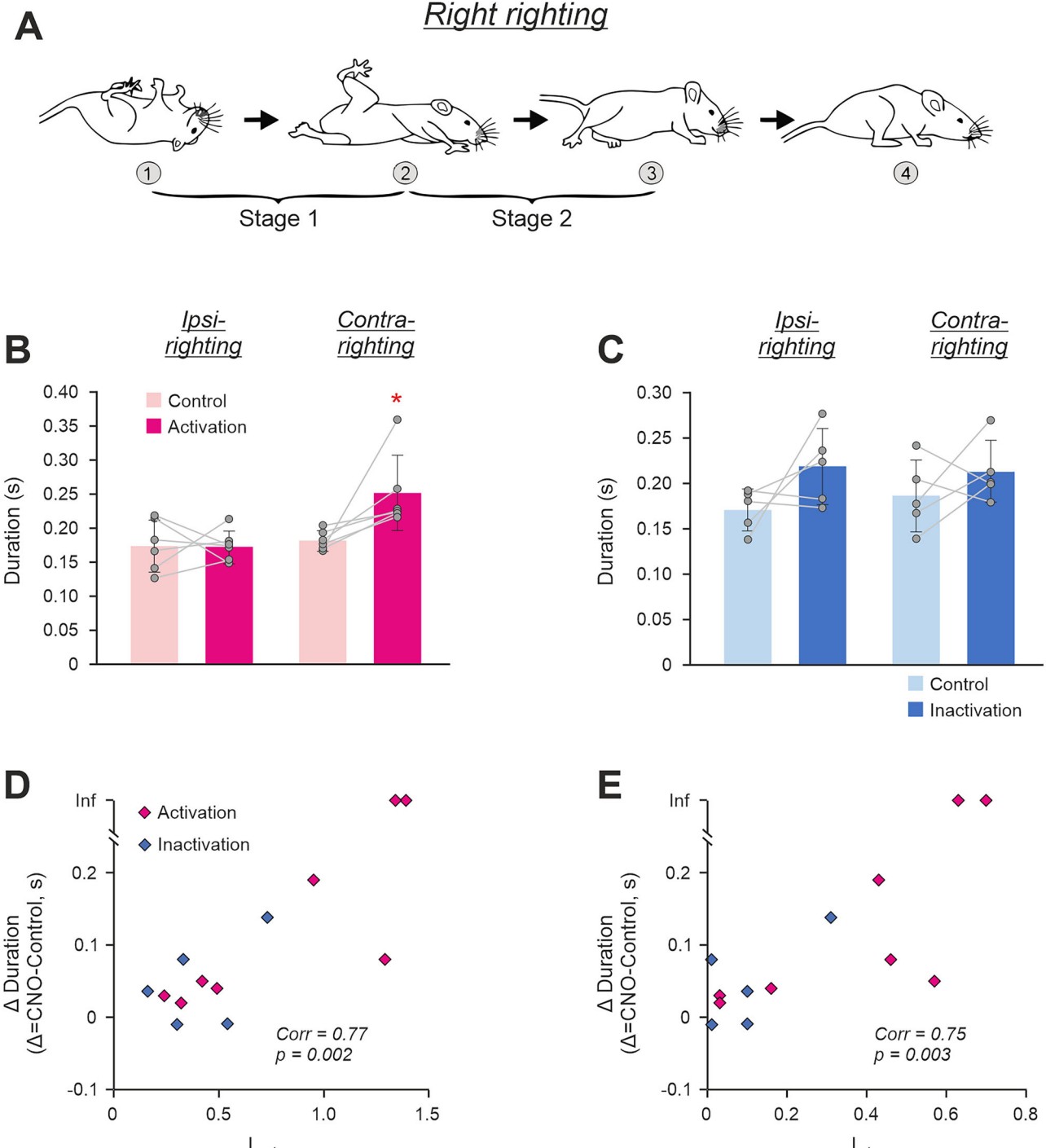

**Fig. 5 | Effects of unilateral activation/inactivation of CaMKIIa-RNs on the righting behavior. A** Sequential positions of a mouse (*1–3*) that, starting from an upside-down position, acquired a dorsal side-up position. Two stages of the righting (*Stage 1* and *Stage 2*) are indicated. **B**, **C** Values of durations of the ipsilateral as well as contralateral righting (*Ipsi*- and *Contra-righting*, respectively) in individual animals, as well as the corresponding mean ± SD values, before (*Control*), during unilateral activation (**B**), and inactivation (**C**) of CaMKIIa-RNs. **D**, **E** Positive correlation between the abduction/adduction asymmetry index ($I_{AB/AD}$ in **D**) as well as

the extension/flexion asymmetry index ($I_{E/F}$ in **E**) during unilateral activation/inactivation of CaMKIIa-RNs and the increase in duration of the righting performed toward the less active subpopulation of CaMKIIa-RNs as compared to control. *Corr*, Spearman's rank correlation coefficient. Two animals that during unilateral activation of CaMKIIa-RNs were unable to perform the contralateral righting are indicated by points with the infinity (*Inf*) ordinate. In **B–D**, and **E**: $N$ = 6, 5, 13, and 13, respectively.

ipsilateral turning (circling) during locomotion in the open field (Fig. 7F, upper panel), while during unilateral activation of CaMKIIa-RNs, animals performed locomotion with right and left turns which randomly occurred in approximately equal proportion (Fig. 7F, lower panel). These differences in effects of unilateral activation of CaMKIIa-RNs and Vglut2-RNs were

observed in all studied animals ($N$ = 7 for CaMKIIa-RNs and $N$ = 3 for Vglut2-RNs).

Second, we compared the effects of unilateral activation of inhibitory GABAergic neurons and CaMKIIa-RNs in MdV-IRt area. For this purpose, in GAD67-Cre mice we infected GAD67 reticular neurons (GAD67-RNs)

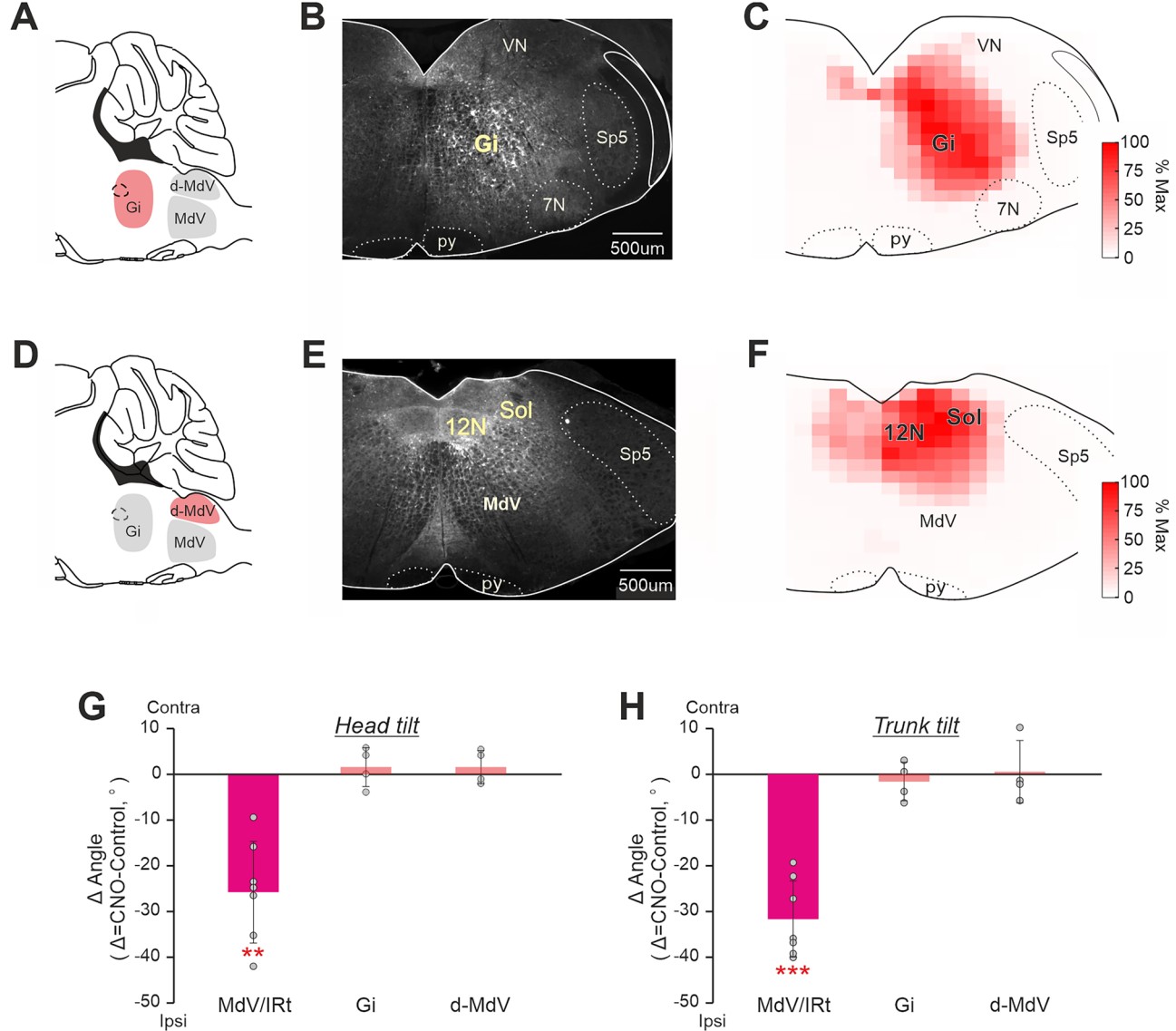

**Fig. 6 | Comparison of effects of unilateral activation of CaMKIIa-RNs located in MdV-IRt area and in two adjacent areas. A, D** A scheme of the sagittal section of the brainstem indicating the location of two areas (the gigantocellular nucleus, Gi, and the area that is dorsal to MdV, d-MdV) adjacent to MdV-IRt area. **B, E** Representative examples of unilateral infection of CaMKIIa-RNs in Gi (**B**) and in d-MdV (**E**) with AAV-CaMKIIa-hM3Dq-mCherry. **C, F** Heatmaps showing the averaged extent of the infected area in Gi (**C**, N = 4) and in d-MdV (**F**, N = 4). **G, H** Comparison of effects of unilateral activation of CaMKIIa-RNs in MdV-IRt, in d-MdV, and in Gi area on the head (**G**) and trunk (**H**) roll tilt in animals standing on a horizontal surface. In **G** and **H**, mean ± SD values of the difference between two conditions (unilateral activation and control) are shown. In **G, H**: N = 7 for MdV-IRt, N = 4 for Gi, and N = 4 for d-MdV.

in MdV-IRt area with AVV-hsyn-DIO-hM3Dq-mCherry (Fig. 8A, B). Figure 8C, D compares effects of unilateral activation of GAD67-RNs and CaMKIIa-RNs in MdV-IRt area. During standing on a horizontal surface, for the roll tilt angles of the head and trunk, differences from the corresponding control values were close to 0 (95% confidence intervals were from 1.4° to 6.9° for the head tilt, and from -1.4° to 3.4° for the trunk tilt) during unilateral activation of the GAD67-RNs. By contrast, the differences during unilateral activation of CaMKIIa-RNs and control was large and statistically significant both for the head and trunk tilts (respectively, –25.6 ± 11.0° and –31.8 ± 8.5°, one-sample $t$ test, $p = 9 \times 10^{-4}$, and $p = 6 \times 10^{-5}$). Thus, left-right asymmetry in activity of GABAergic neurons in MdV-IRt area does not affect the body orientation in the transverse plane.

Taken together, all these results suggest that the change of the body orientation in the transverse plane (the body roll tilt) is caused by left-right asymmetry in activity of a specific molecularly identified population of RNs

located in a definite brainstem area. It is a population of RNs expressing CaMKIIa and located specifically in MdV-IRt area.

**The majority of CaMKIIa-RNs in MdV-IRt area are glutamatergic**
To reveal neurotransmitter phenotypes of CaMKIIa-RNs located in MdV-IRt area, first, AAV-CaMKIIa-GFP viruses were injected in MdV-IRt area to label CaMKIIa-RNs with GFP (upper panels in Fig. 9A, B). Then by using RNAscope in situ hybridization for vesicular glutamate transporter 2 (Vglut2) and vesicular inhibitory amino acid transporter (Vgat), Vglut2 positive and Vgat positive neurons in MdV-IRt area were identified (middle panels in Fig. 9A, B). We found both neurons with co-expression of GFP and Vglut2 (lower panel in Fig. 9A), as well as neurons with co-expression of GFP and Vgat (lower panel in Fig. 9B) in MdV-IRt area. Thus, the population of CaMKIIa-RNs contains both excitatory glutamatergic and inhibitory GABAergic/glycinergic neurons. However, the relative number of

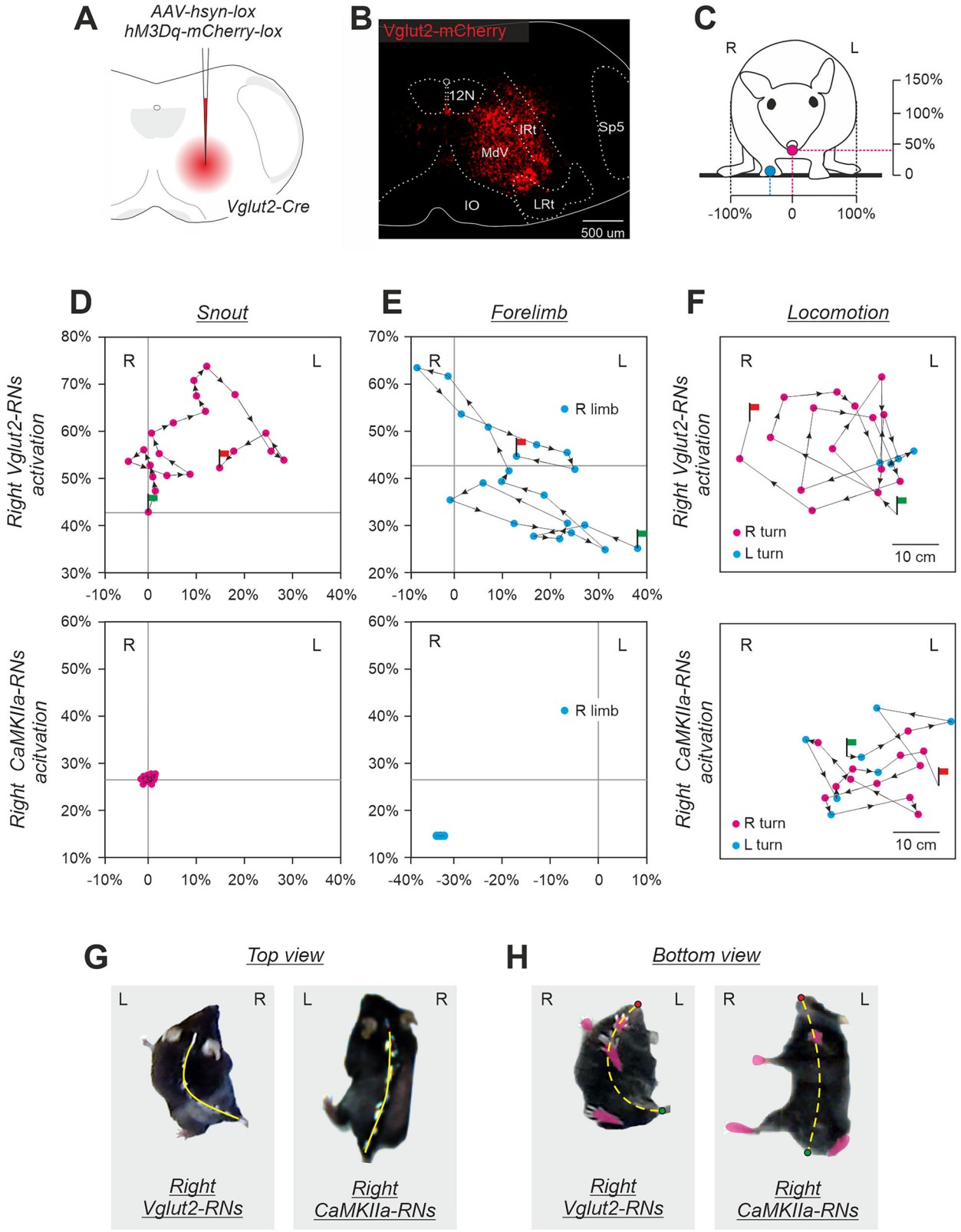

excitatory CaMKIIa-RNs was significantly (more than twofold) higher than the relative number of the inhibitory ones (respectively, 61% *vs* 26%, $\chi^2$ test, $p < 1 \times 10^{-5}$; Fig. 9E). Both glutamatergic and inhibitory CaMKIIa-RNs were distributed within the whole MdV-IRt area. However, the density of glutamatergic neurons was higher in more medial zone (Fig. 9C), while the density of inhibitory neurons was higher in more lateral zone (Fig. 9D).

## Population of CaMKIIa-RNs located in MdV-IRt area contains reticulospinal neurons

To determine whether the population of CaMKIIa-RNs in MdV-IRt area contains reticulospinal neurons, we examined presence of the mCherry signals in the spinal cord sections of mice with unilateral injection of AAV-CaMKIIa-hM3Dq-mCherry in MdV-IRt area. Fig. 10A–F shows a

**Fig. 7 | Comparison of effects of unilateral activation of CaMKIIa-RNs and *Vglut2*-RNs located in MdV-IRt area. A, B** Unilateral infection of Vglut2-RNs with AAV-hsyn-DIO-hM3Dq-mCherry in the Vglut2-Cre mouse. **C** The front view of a mouse. Crimson and cyan circles indicate the positions of the snout and ipsilateral forelimb, respectively. The horizontal and vertical axes for positions of the snout (**D**) and the forelimb (**E**) are shown. The right and left edges of the body correspond, respectively, to –100% and +100% of the half of the body width; 0 for the vertical axis is at the support surface; the vertical scale is the same as the horizontal one. **D, E** Comparison of trajectories of the snout (**D**) and the right forelimb (**E**) movements in the transverse plane in the animal with unilateral activation of the right Vglut2-RNs (upper panels) and in the animals with unilateral activation of the right CaMKIIa-RNs (lower panels); 0,02 s between points. **F** Comparison of trajectories of locomotion in the open field performed by the mouse during activation of the right Vglut2-RNs and by the mouse during activation of the right CaMKIIa-RNs; 1 s between points. In **C–F**, Green and red flag indicate the beginning and the end of the trajectory, respectively. **G, H** Top and bottom views of the mouse during activation of the right Vglut2-RNs and mouse during activation of the right CaMKIIa-RNs. In **G** and **H**, the position of the spine and the midline of the body are shown by solid and dashed yellow lines, respectively.

**Fig. 8 | Comparison of effects of unilateral activation of CaMKIIa-RNs and *GAD67*-RNs located in MdV-IRt area. A, B** Unilateral infection of GAD67-RNs with AAV-hsyn-DIO-hM3Dq-mCherry- in the *GAD67*-Cre mouse. **C, D** Comparison of effects on the head (**C**) and trunk (**D**) roll tilt caused by unilateral activation of GAD67-RNs. Values of averaged difference between the roll tilt angle observed during unilateral activation of neurons and in control (*Δ Angle*) are shown for individual animals as well as corresponding mean ± SD values. In **C, D**: *N* = 3 for *GAD67*-RNs and *N* = 7 for CaMKIIa-RNs.

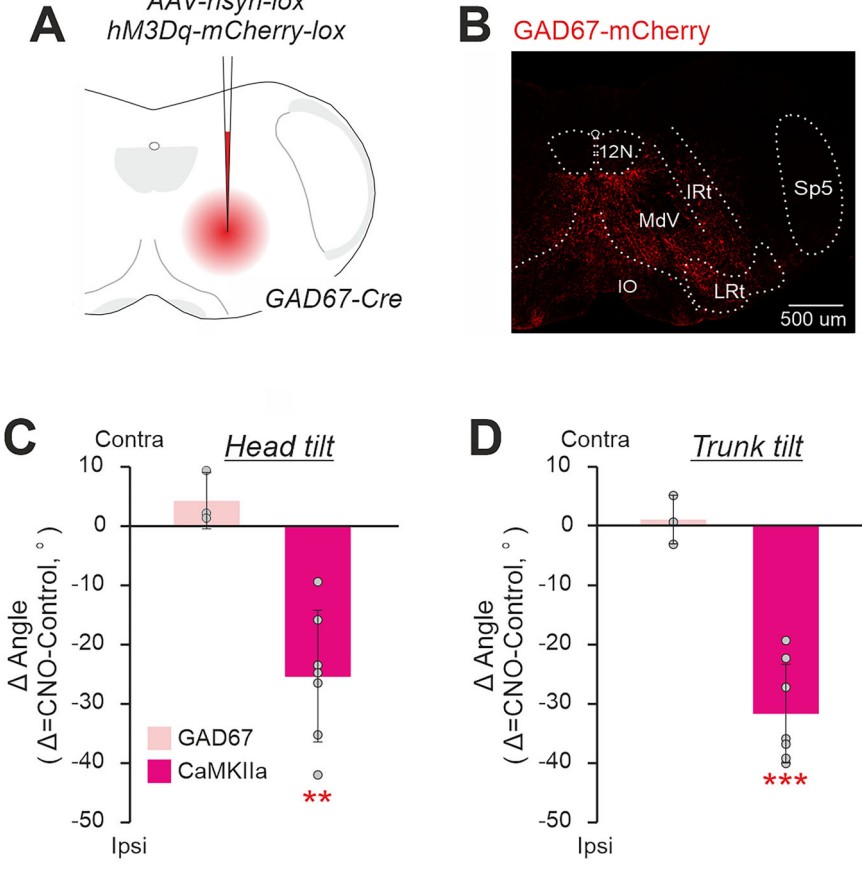

representative example illustrating position of mCherry⁺ axons and their arborizations in the gray matter at the cervical (**A, D**), thoracic (**B, E**), and lumbar (**C, F**) levels of the spinal cord. We found mCherry⁺ axons at all these spinal levels suggesting that the population of CaMKIIa-RNs in MdV-IRt area contains reticulospinal neurons. In the spinal cord, mCherry⁺ axons descended mainly in the medial part of the ipsilateral lateral funiculus and their number decreased from cervical to lumbar level (Fig. 10A–F). At all three levels of the spinal cord, the greatest arborization of mCherry⁺ axons was found within the intermediate part of the ipsilateral gray matter in laminae VII, VIII and X (Fig. 10G–I). Also, at the cervical and thoracic levels a substantial arborization was found in the same laminae of the contralateral gray matter (Fig. 10A, B, G, H). Notably, mCherry⁺ terminals were largely absent in the dorsal horn laminae I–VI where sensory networks are localized, and in lamina IX where the majority of limb muscle motoneurons reside. Since lamine VII and VIII contain interneurons and laminae X contains motoneurons of the axial muscles, one can conclude that CaMKIIa-reticulospinal neurons most likely directly affect predominantly interneurons as well as motoneurons of the axial muscles at all levels of the spinal cord.

## Discussion

In the present study, we identified a population of reticular neurons located in the caudal medulla—CaMKIIa-RNs in MdV-IRt area—which control body orientation maintained by an animal in the transverse plane.

Although it was reported that the caudal medulla contains rather sparse CaMKIIa neurons[33,40], we demonstrated that unilateral chemogenetic activation or inactivation of the CaMKIIa-RNs located in the MdV-IRt area elicited a lateral body sway towards the dominant (more active) side, and this new body orientation was actively stabilized during standing and maintained during locomotion. Previously, we showed a similar effect (active stabilization of body roll tilt) caused by binaural galvanic vestibular stimulation (GVS)[17,18]. We suggest that the mechanisms behind the induction of stabilized body roll tilt caused by asymmetry in activity of CaMKIIa-RNs and by asymmetry in vestibular inputs evoked by GVS are similar. Previous studies have suggested that terrestrial quadrupeds stabilize body orientation in the transverse plane due to continuous interaction of antagonistic postural limb reflexes (PLRs) generated by the left and right limbs[17,18] (Fig. 11A). The animal stabilizes such orientation at which the effects of antagonistic PLRs are equal to each other. This is a set-point of the postural control system —the dorsal side-up body orientation that the animal actively stabilizes. Any deviation from this orientation leads to an enhancement of PLRs generated by the left or right limbs, which return the body to the initial orientation (Fig. 11A). We suggest that left-right asymmetry in the activity of CaMKIIa-RNs (or in the activity of vestibular afferents) evokes a shift of the set-point of the postural system through a change of gains in antagonistic PLRs (Fig. 11C, D). This leads to stabilization of a new

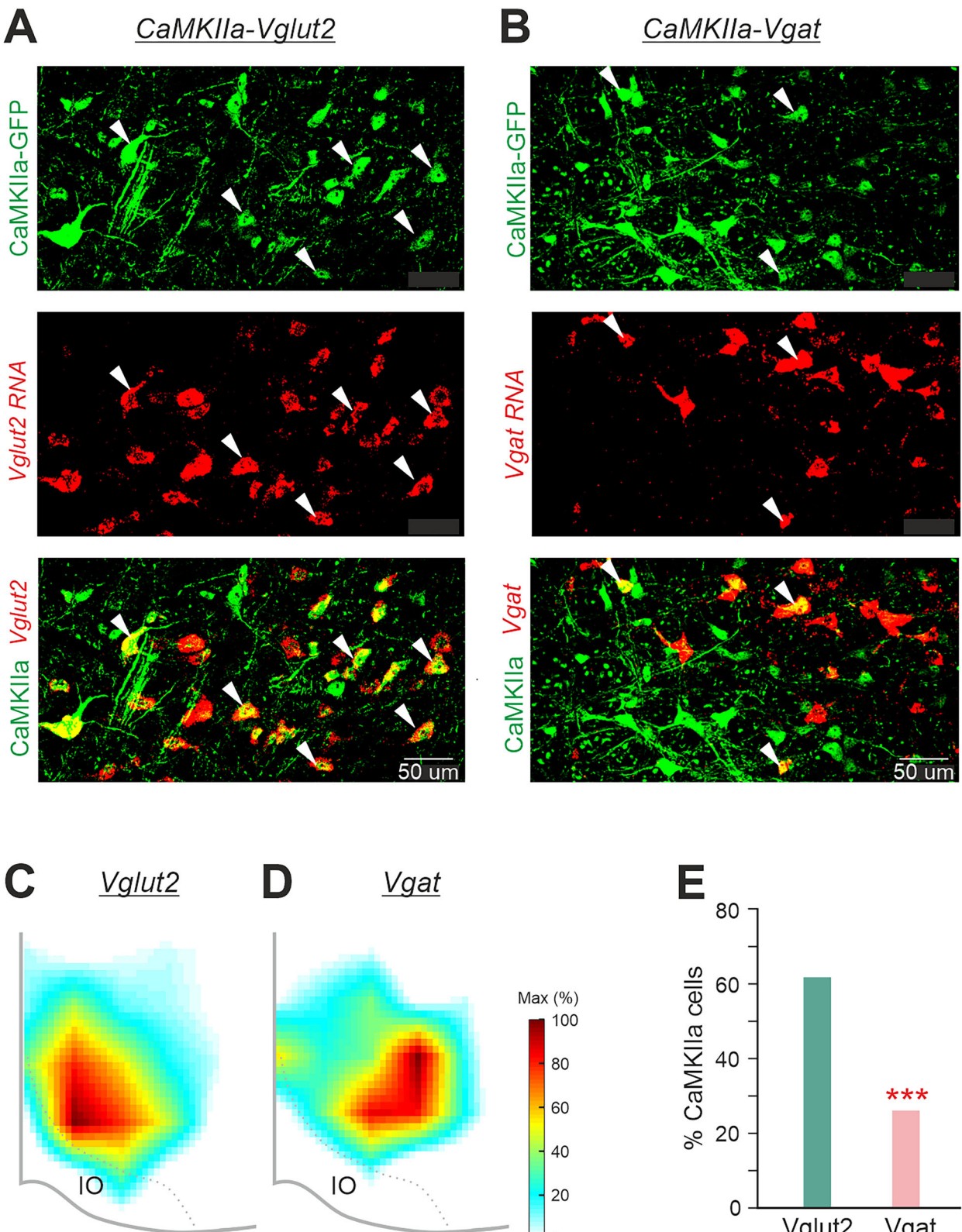

**Fig. 9 | Neurotransmitter phenotypes of CaMKIIa-RNs located in MdV-IRt area.**
**A**, **B** Two representative images with GFP expression in CaMKIIa-RNs (green cells
in the upper panels), with in situ hybridization for Vglut2 mRNA identifying glu-
tamatergic RNs (**A**, red cells in the middle panel), with in situ hybridization for Vgat
mRNA identifying inhibitory RNs (**B**, red cells in the middle panel), with co-
expression of GFP and for Vglut2 mRNA identifying glutamatergic CaMKIIa-RNs
(**A**, yellow cells in the lower panel), and with co-expression of GFP and for Vgat
mRNA identifying inhibitory CaMKIIa-RNs (**B**, yellow cells in the lower panel).
White arrowheads in (**A**) and **B** mark glutamatergic and inhibitory CaMKIIa-RNs,
respectively. **C**, **D** Heatmaps showing density of glutamatergic (Vglut2) (**C**) and
inhibitory (*Vgat*) (**D**) CaMKIIa-RNs in MdV-IRt area. **E** Percentage of Vglut2 and
*Vgat* positive CaMKIIa-RNs in MdV-IRt area. For identification of Vglut2
CaMKIIa-RNs and *Vgat* CaMKIIa-RNs: $N = 3$, $n = 6$ sections, 717 neurons, and
$N = 3$, $n = 6$ sections, 287 neurons, respectively.

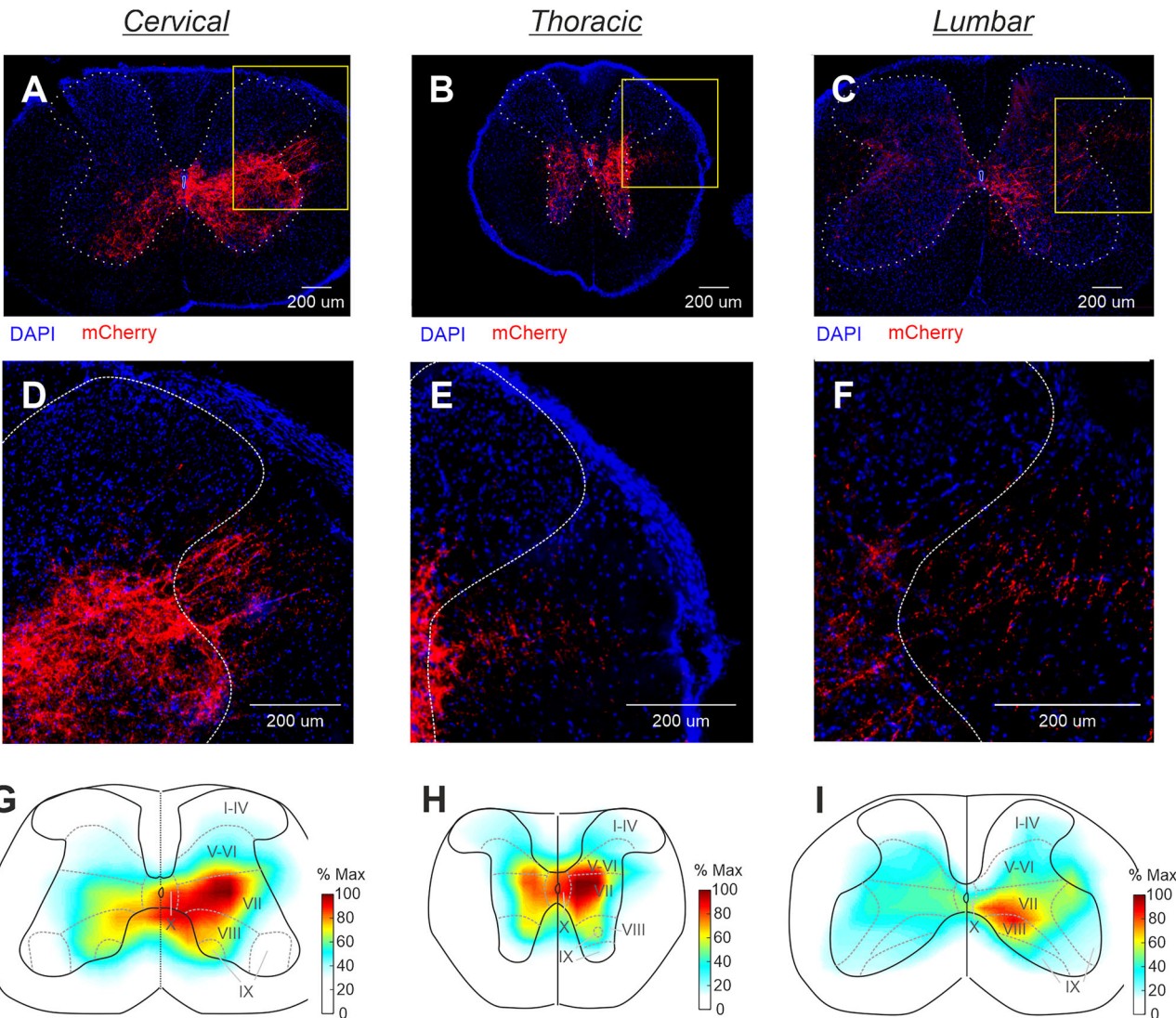

**Fig. 10 | Population of CaMKIIa-RNs in MdV-IRt area contains reticulospinal neurons. A–F** A representative example of mCherry⁺ axons located mainly in the medial part of the ipsilateral lateral funiculus, as well as their arborizations in the gray matter at the cervical (**A**, **D**), thoracic (**B**, **E**), and lumbar (**C**, **F**) levels of the spinal cord. Yellow rectangles in (**A–C**) delineate the areas shown at higher magnification in (**D–F**), respectively. These insets illustrate position of axons of CaMKIIa reticulospinal neurons in the ipsilateral lateral funiculus at cervical (**D**), thoracic (**E**), and lumbar (**F**) levels. **G–I** Heatmaps showing averaged fluorescent intensity (% Max) of mCherry⁺ axons and their arborizations in the cervical (**G**), thoracic (**H**), and lumbar (**I**) spinal cord (N = 2, n = 10, 12, and 18, for cervical, thoracic, and lumbar spinal cord, respectively).

orientation of the body with some roll tilt. Asymmetry in the activity of CaMKIIa-RNs evokes roll tilt toward the dominant side (Fig. 11C, D). In the natural environment, such a shift in the set-point of the postural system allows the animal to stabilize the dorsal side-up body orientation on a laterally inclined surface (Fig. 11E). A similar principle of balance control was also found in simpler animals—a mollusk (Clione) and a lower vertebrate (lamprey)[15,34,41]. In the lamprey, antagonistic postural reflexes are mediated by two populations of reticulospinal neurons, and different factors that produce asymmetry in their activity, affect the body orientation stabilized in the transverse plane. To clarify whether left-right asymmetry in the tonic activity of CaMKIIa-RNs changes the gain of PLRs that leads to stabilization of a new orientation in the transverse plane, or whether PLRs in mice are mediated by CaMKIIa-RNs, is a question for future studies. It should be noted that while hM3Dq and hM4Di are designed to increase or reduce excitability of neurons, it remains unclear whether these neurons retain the ability to respond to inputs after CNO injection or whether they reach their maximal activity or become fully inhibited.

Since reticulospinal neurons receive substantial vestibular input[21,22], one can assume that left/right asymmetry in the activity of reticulospinal neurons contributes to the shift in the set-point of the postural system caused by GVS. Whether CaMKIIa-RNs in the MdV-IRt area specifically receive vestibular input, and thus potentially can contribute to the change in body orientation caused by GVS remains unknown. However, the asymmetry in activity of CaMKIIa-RNs can be evoked by other types of sensory information, e.g., on the basis of the visual information about a laterally tilted surface in front of the animal. Inputs from multiple upstream regions (motor cortex, superior colliculus, red nucleus, cerebellar deep nuclei, and other brainstem reticular nuclei) to glutamatergic MdV neurons have been demonstrated[42].

The fact that not only unilateral activation but also unilateral inactivation of CaMKIIa-RNs creates an asymmetry in their activity sufficient for the behavioral effects suggests that the population of CaMKIIa-RNs in the MdV-IRt area has a substantial level of activity in standing animals. This finding supports our suggestion about their important role in the control of body orientation. However, unilateral inactivation of CaMKIIa-RNs in

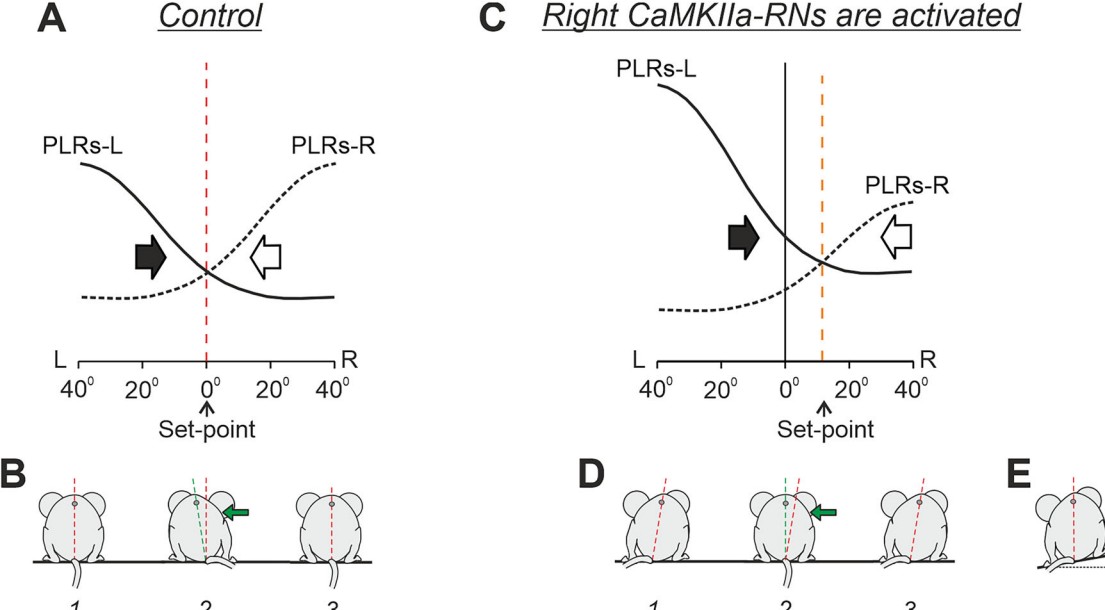

**Fig. 11 | Conceptual model of the trunk stabilization system and effects of left-right asymmetry in activity of CaMKIIa-RNs. A–D** Presumed effects of the two antagonistic postural limb reflexes (PLRs) in the unrestrained standing mouse in control (**A** and **B**) and during activation of right CaMKIIa-RNs (**C** and **D**). **A** and **C**: the abscissa shows a deviation of the dorso-ventral body axis from the vertical (lateral sway); the ordinate shows the values of the right and left (L) PLRs (PLR-R and PLR-

L, solid and dashed lines, respectively). Black and white arrows indicate the motor effect (lateral sway) caused by PLR-R and PLR-L, respectively. **B**, **D** The stabilized orientation (*1*), effect of the lateral push (*2*), and the restored orientation (*3*). **E** The shift of the set-point of the postural control system caused by the left-right asymmetry in activity of CaMKIIa-RNs allows to the animal stabilize the dorsal side-up orientation on the laterally inclined surface. (See DISCUSSION for details).

standing animals caused much weaker behavioral effects compared to those caused by unilateral activation. In particular, it was reflected in much smaller absolute values of the head and trunk roll tilts. Also, unilateral inactivation does not significantly affect the duration of contralateral righting, while unilateral activation caused a significant increase in its duration. One can suggest that during quiet standing or during righting, the CaMKIIa-MdV neurons have a relatively low level of activity and thus their unilateral activation and inactivation produce, respectively, strong and weak left-right asymmetries in their activity, leading to correspondingly strong and weak behavioral effects.

By contrast, during locomotion, unilateral inactivation of CaMKIIa-RNs evoked a significant body roll tilt, while the effect of unilateral activation on average was non-significant (although strong in some individual animals). One can suggest that during locomotion, CaMKIIa-RNs are strongly activated. Thus, their unilateral inhibition creates a strong left-right asymmetry in the activity of the population, leading to a pronounced behavioral effect. By contrast, their unilateral activation creates a very weak (if any) asymmetry in the population activity (since the neuronal activity is already near its maximal level) which results in the absence of a behavioral effect.

We did not find asymmetry in the timing of locomotor movements performed by the left and right limbs during unilateral activation or unilateral inactivation of CaMKIIa-RNs. However, we found that unilateral inactivation evoked a significant increase in the cycle duration due to an increase in stance duration, while unilateral activation did not. These results further confirm that during locomotion, effects of unilateral inactivation are stronger than those of unilateral activation. The reason for the decrease in locomotor speed is not clear. It could be through an effect of asymmetrical activity of CaMKIIa-RNs on the rhythm generating components of the locomotor CPGs, or a result of postural asymmetry making it more difficult to maintain balance during locomotion on a horizontal surface.

We showed that unilateral activation/inactivation of CaMKIIa-RNs located in the MdV-IRt area evoked flexion and adduction of the limbs on the dominant side, and simultaneous extension and abduction of the opposite limbs. These results are in line with the results of an earlier study that demonstrated that electrical microstimulation of specific sites in the

pontomedullar reticular formation evoked inactivation and activation of extensors in the ipsilateral and contralateral hindlimb, respectively[23].

It was previously demonstrated that asymmetry in configurations of the left and right limbs (flexion of limbs on one side and extension of the opposite limbs) evoked body roll tilt, observed in context of postural corrections caused by the lateral tilt of the supporting surface[17,34,35]. We suggest that asymmetry in limb configurations (flexion and adduction on the dominant side and simultaneous extension and abduction on the opposite side), caused by asymmetry in the activity of CaMKIIa-RNs, led to the observed body roll tilt in our study. However, the contribution of the trunk muscles to the execution of the body roll tilt cannot be ruled out.

We demonstrated that the behavioral effect (the body roll tilt) evoked by the left-right asymmetry in activity of CaMKIIa-RNs is specific to the population of CaMKIIa-RNs located in MdV-IRt area, but not in the neighboring areas. Since the same viral titer and injection volume were used in experiments with virus injection in MdV-IRt area and in neighboring areas, the differences in behavioral effects are unlikely to have been caused by differences in viral spread. It should be noted that in our experiments, some viral spread to the inferior olive (IO) was observed due to the proximity between IO and MdV. Previous studies have shown that widespread lesions or genetic silencing of IO neurons can induce dystonia-like symptoms and affect posture[43,44]. However, in our experiments, the extent and location of IO infection varied across animals and did not correlate with the magnitude of the head/trunk roll tilt. By contrast, consistent infection of the MdV-IRt area reliably produced the body roll tilt. Thus, most likely CaMKIIa-RNs, rather than IO neurons, are the primary drivers of the postural effects.

Furthermore, within the MdV-IRt area itself, the body roll tilt is evoked specifically by CaMKIIa-RNs, but not by the broader glutamatergic (Vglut2-RNs) or GABAergic (GAD67-RNs) populations. We found that the left-right asymmetry in activity of Vglut2-RNs located in MdV-IRt area evoked another behavioral effect, a body bend in the yaw plane toward the dominant side accompanied by continuous movements of the head and the ipsilateral forelimb. Since we demonstrated that the majority of CaMKIIa-RNs are Vglut2 neurons, the activated Vglut2-RN population contained

CaMKIIa-RNs. Most likely, during activation of all Vglut2-RNs in MdV-IRt, the effect of CaMKIIa-RNs on the body orientation in the transverse plane was hidden by the strong behavioral effects of non-CaMKIIa Vglut2 neurons (the body bending in the yaw plane). Previously, it was demonstrated that ablation of Vglut2-RNs located in MdV distorts skilled forelimb motor tasks[42] and also affects orofacial movements[45]. It was demonstrated that both the Vglut2 and inhibitory neurons in IRt contribute to rhythmic orofacial motor behaviors, such as whisking and licking[45]. Thus, the MdV-IRt area of the caudal medulla contains a number of different molecularly identified populations of RNs that control specific aspects of the motor behavior in mice.

Some of the effects observed in the present study during unilateral activation of Vglut2-RNs in MdV-IRt (lateral bending of the trunk and circling during locomotion) are similar to those caused by unilateral activation of V2a neurons in the gigantocellular reticular nucleus[28,29]. Thus, glutamatergic neurons located in different reticular nuclei contribute to the control of body configuration in the yaw plane.

We found that the majority of neurons in the population of the CaMKIIa-RNs in MdV-IRt area are excitatory glutamatergic (Vglut2) neurons although it also contains inhibitory GABAergic/glycinergic (Vgat) neurons. These results are in line with the results of previous studies that documented that MdV contains both glutamatergic and inhibitory neurons[42], and that CaMKIIa neurons can be excitatory as well as inhibitory[33,46]. A goal of future studies will be to elucidate whether the evoked roll tilt of the body is due to left-right asymmetry in activity of only excitatory CaMKIIa-RNs, or only inhibitory CaMKIIa-RNs, or both excitatory and inhibitory CaMKIIa-RNs.

We demonstrated that the population of CaMKIIa-RNs located in MdV-IRt contains reticulospinal neurons with axons descending ipsilaterally and branching at all levels of the spinal cord with the intensity of branching in the cervical region higher than in the lumbar region. Reticulospinal neurons originating from MdV with similar projections were described earlier[42,47]. However, while reticulospinal neurons originating from MdV formed abundant synaptic connections with limb motorneurons[42,47], terminals of the population of CaMKIIa reticulospinal neurons originating from MdV-IRt area avoid lamina IX where the majority of limb motoneurons reside, suggesting that their effects on limb motoneurons are mediated by spinal interneurons. A recent study demonstrated a significant input from the MdV neurons to spinal V1 interneurons, further supporting the suggestion that projections from this region target spinal interneurons[48]. V1 interneurons are known to modulate locomotion, including locomotor frequency and timing of flexor versus extensor activity[49–51]. Since activation or inactivation of CaMKIIa neurons similarly alters cycle and stance durations, this reinforces the possibility that CaMKIIa neurons may project onto this specific interneuron population.

A question for future studies is whether the body roll tilt is evoked by the left-right asymmetry in activity of only CaMKIIa reticulospinal neurons, or only CaMKIIa-RNs that do not project to the spinal cord, or the entire population of CaMKIIa-RNs. In the case that CaMKIIa reticulospinal neurons contribute to control of the body roll tilt, the fact that one of the areas of arborization of their axons coincides with the area where motoneurons of axial muscles are located (laminae X) supports the assumption about the contribution of trunk muscles to the execution of the body roll tilt. Also, in the case that reticulospinal CaMKIIa neurons play a crucial role in induction of the behavioral effects, one of possible explanations of strong effect on hindlimbs despite denser axonal branching of CaMKIIa reticulospinal axons in cervical region, is that the effects are mediated by propriospinal interneurons receiving input from CaMKIIa reticulospinal neurons terminating in cervical and thoracic segments.

It is well documented that at the acute stage of incomplete spinal cord injury, motor functions (including postural functions) are severely distorted or absent. However, they gradually recover over time due to plastic changes in the corresponding neuronal networks[6,52–57]. It was shown that reticular neurons, and specifically glutamatergic neurons in the medullary reticular formation, contribute to these plastic changes[58–61]. In the present study, we demonstrated that CaMKIIa-RNs in MdV-IRt area are involved in the control of posture. Since calcium–calmodulin-dependent protein kinase IIa is involved in synaptic plasticity[31,62], one can expect that CaMKIIa-RNs from MdV-IRt area may contribute to plastic changes underlying the recovery of postural functions after the spinal cord injury.

To conclude, in the present study, a population of CaMKIIa-RNs in the MdV-IRt area was characterized and its functional role was demonstrated. We found that left-right asymmetry in activity of this population evoked a body roll tilt which was actively stabilized during standing and maintained during locomotion. We suggest that CaMKIIa-RNs in MdV-IRt area control the body orientation in the transverse plane. To maintain the dorsal-side-up body orientation during standing on a horizontal surface, the activity level of the right and left CaMKIIa-RNs must be equal to each other. On the other hand, to maintain the dorsal-side-up orientation on a laterally inclined surface, a right/left asymmetry in the CaMKIIa-RNs activity is necessary. We found that most CaMKIIa-RNs are excitatory, and that the population contains reticulospinal neurons. The obtained results advance our understanding of the neuronal mechanisms underlying stabilization of the body orientation at different environmental conditions.

## Methods
### Animals
Experiments were performed on wild type (C57BL6, $N = 24$: 14 females and 10 males) mice, as well as on Vglut2-Cre ($N = 3$) and GAD67-Cre ($N = 3$) transgenic mice. The mice were of both sex, 18–30 g weight, and 8–12 weeks old at the time of virus injection. They were housed in standard cages with food and water ad libitum at a 12 h light/12 h dark cycle. All experiments were conducted with approval of the local ethical committee (Norra Djurförsöksetiska Nämnden) in Stockholm and followed the European Community Council Directive (2010/63EU) and the guidelines of the National Institute of Health Guide for the Care and Use of Laboratory animals. We have complied with all relevant ethical regulations for animal use. No experimental criteria were set. No unexpected adverse events.

### Stereotaxic viral injections
All surgical procedures were performed under general anesthesia and aseptic conditions. General anesthesia consisted of ketamine (75 mg/kg) in combination with medetomidine (1 mg/kg) administered intraperitoneally. The level of anesthesia was controlled by applying pressure to a paw (to detect limb withdrawal), and by examining the size and reactivity of pupils. Anesthetized mice were fixed in a stereotaxic frame. Mice were kept on a 37 °C heating pad for the duration of the surgery. Viscotears was used for lubrication of the eyes to prevent dehydration. The skin on the head was shaved, and an incision was made to expose the skull. Skull references were taken for bregma and lambda. A small hole was drilled in the skull overlying the target brain region for injection. The underlying dura was opened. A pulled glass capillary was filled with mineral oil, secured to a capillary nanoinjector that was fixed on a micromanipulator. Viruses were mixed with a small amount of Fast Blue for visualization and loaded into the capillary. The capillary was advanced to the target brain region at a rate of 0.1 mm/s, and injection was performed at a rate of 100 nl/min. In total, 300 nl was injected in one site. The capillary was left in place for 5 min following the injection and then withdrawn at a rate of 0.1 mm/s. For analgesia, buprenorphine (0.05–0.1 mg/kg) was given subcutaneously postoperatively and twice daily for the next 2 days. Humane endpoints were defined in the ethical protocol; animals were monitored daily for signs of distress. Experiments started 3–6 weeks after surgery.

For manipulations with the activity of CaMKIIa-RNs, we used a chemogenetic approach. For activation and inactivation of CaMKIIa-RNs in the caudal medulla (the area of the medullary reticular nucleus ventral part and the intermediate reticular nucleus, MdV-IRt area), respectively, AAV5-CaMKIIa-hM3Dq-mCherry-WPRE and AAV5-CaMKIIa-hM4Di-mCherry-WPRE ($5 \times 10^{12}$ vg/ml, volume 300 nl, Viral Vector Facility, University of Zurich, v.96 and v.102) were injected in wild type mice unilaterally with the following coordinates: -7.4 mm antero-posterior from

bregma, 0.7 mm lateral, and 5.6 mm ventral. To confirm specificity of effects caused by manipulation with activity of CaMKIIa-RNs in this area, AAV5-hsyn-DIO-hM3Dq-mCherry-WPRE ($4 \times 10^{12}$ vg/ml, Viral Vector Facility, University of Zurich, v.89) was unilaterally injected to the same area in Vglut2-Cre and GAD67-Cre mice to target all glutamatergic and GABAergic neurons, respectively. To check that the effects of manipulation with activity of CaMKIIa-RNs were specific for MdV-IRt area, AAV5-CaMKIIa-hM3Dq-mCherry-WPRE ($5 \times 10^{12}$ vg/ml, volume 300 nl) was unilaterally injected in rostral and dorsal adjacent areas in wildtype mice (Fig. 6). The following coordinates for injections to the adjacent areas were used: –6 mm antero-posterior from bregma, 0.7 mm lateral and 5.6 mm ventral, for targeting the dorsal adjacent area; –7.4 mm antero-posterior from bregma, 0.7 mm lateral, and 4.5 mm ventral for targeting the rostral adjacent area. To identify the neurotransmitter type of CaMKIIa-RNs, labeling of CaMKIIa-RNs was combined with RNAscope. For this purpose, we injected 300 nl AAV5-CaMKIIa-EGFP-Cre-WPRE ($6 \times 10^{12}$ vg/ml, Viral Vector Facility, University of Zurich, v.315) into wild type mice.

## Experimental designs

For activation/inactivation of infected neurons, CNO (Tocris, catalog no. 4936) dissolved in DMSO was administered intraperitoneally at a dose of 1 mg/kg. Mice performed each of four basic motor behaviors (standing on a horizontal surface, postural corrections on a tilting platform, forward locomotion, and righting) before and between 40–90 min after injection of CNO. Animals were acclimatized to the apparatus for the day before testing. Experimental designs used for the behavioral experiments were described earlier[36,63] and are presented here in brief.

To analyze the basic body orientation and configuration during standing on a horizontal surface, the animal was positioned on a transparent tilting platform ($12 \times 12$ cm) that was oriented horizontally (Fig. 2A, D). To evoke postural corrections, the platform with the animal, whose sagittal plane was aligned to the axis of the platform rotation, was tilted periodically in the frontal (transverse) plane of the animal (roll tilt α) with the amplitude of ±20° (Fig. 3A, B). A trapezoid tilt trajectory with the transitions between extreme positions lasting for ~0.5–1 s, and each position maintained for ~1–1.5 s was used. It was necessary to habituate animals to the tilting platform and to train them to stand still during tilts. For this purpose, the animal was positioned on the tilting platform, and tilts with increasing amplitude were applied. If the animal started to walk, it was returned to the initial standing position by the experimenter. Usually, a 20 min session of such training performed during 2–3 days was sufficient to evoke episodes, in which the mouse maintained the standing posture with its sagittal plane aligned to the axis of the platform rotation leading to generation of postural corrections in response to 5–7 sequential tilt cycles.

Forward locomotion was performed in a corridor setup. The setup consisted of a corridor (length 50 cm, height 4 cm, width 2.5–3.5 cm) with a small box ($7 \times 7 \times 4$ cm) at each end of the corridor. Each box had a removable top and a door that closed the entrance to the corridor. The animal was placed in the entrance box through the removable top, the top was closed, then the doors to the corridor of both boxes were opened and the animal could easily walk in the corridor straight forward but could not turn around. When the mouse entered the box on another side of the corridor, it turned and performed forward locomotion in the opposite direction. Usually, the animal spontaneously exhibited 3–4 sequential episodes of forward locomotion. Each episode of forward walking in the corridor consisted of 5–9 steps.

To evoke righting behavior (Fig. 5A), a mouse was positioned on its back (with its ventral side up) on a horizontal surface. Then the animal was released so it could assume the normal body orientation characteristic for standing.

## Recording and data analysis

The hindlimbs and trunk of the animal were shaved, and markers were drawn on the skin along the spine. The video camera was positioned at a distance of ~2 m from the mouse. To characterize the kinematics of fast movements performed during locomotion and righting, high-speed video recording (100 frames/s, SIMI motion 9.1.1) was used. Standing on a horizontal surface as well as relatively slow postural corrections were video recorded with a lower speed (50 frames/s). We recorded the top view and bottom view during locomotion, the side view during righting, the rear/front view during standing on a horizontal platform, and the rear view during postural corrections on the tilting platform. During standing, simultaneously with recording of the rear/front view, the view from below was recorded (by means of a 45° tilted mirror positioned under the transparent surface). The video recordings were analyzed off-line frame by frame.

In the following text, terms "ipsilateral" and "contralateral" indicate the side of the mouse, respectively, ipsilateral and contralateral to the side of the virus injection.

The head and trunk orientation in the transverse plane (the roll tilts) were characterized by the angles between the vertical (solid yellow lines in Figs. 2A,D,3A,B) and the dorso-ventral axis of the head or the trunk, correspondingly (red dashed lines in Figs. 2A,D,3A,B). The dorso-ventral axis of the trunk was estimated from the rear view as a line connecting the marker on the spine rostral to the pelvis with the base of the tail. The dorso-ventral axis of the head was estimated from the frontal view as a line perpendicular to the line connecting the eyes. The ipsilateral and contralateral roll tilt angles had positive and negative values, respectively.

Asymmetry in extension/flexion of the hindlimbs during standing on the horizontal surface and during postural corrections was estimated from the rear view and characterized by the extension/flexion asymmetry index: $I_{E/F} = (L_{CONTRA} - L_{IPSI})/(L_{CONTRA} + L_{IPSI})$, where $L_{IPSI}$ and $L_{CONTRA}$ were, respectively, the ipsilateral and the contralateral limb lengths. The limb length was estimated by the distance from the heel to the marker on the spine rostral to pelvis (illustrated in Figs. 2A, 3F). Thus, $I_{E/F} = 0$ if the lengths of the left and right limbs were equal, $I_{E/F} > 0$ if the contralateral limb was more extended than the ipsilateral one, and $I_{E/F} < 0$ if the ipsilateral limb was more extended than the contralateral one.

To characterize the position of the limbs in relation to the trunk (abduction/adduction) during standing on horizontal surface, the view from below was used. The trunk outline and its midline were drawn. Then the axes perpendicular to the midline at the level of forelimbs (one-third of the distance from the nose to the base of the tail) and hindlimbs (two-thirds of the distance from the nose to the base of the tail) were drawn. The midpoint of the body width at the corresponding level was taken as "0". The position of a limb in relation to the trunk was characterized by the coordinate along the corresponding axis ($b$ in Fig. 2G). This coordinate was termed "the lateral position" of the limb and expressed in percent of the body half-width. Positive and negative values of the lateral position of the limb indicated ipsilateral and contralateral location of the foot in relation to the midline, respectively (Fig. 2G–I).

Asymmetry in abduction/adduction of the fore- and hindlimbs was characterized by the abduction/adduction asymmetry index: $I_{AB/AD} = (b_{CONTRA} - b_{IPSI})/(b_{CONTRA} + b_{IPSI})$, where $b_{CONTRA}$ and $b_{IPSI}$ were the lateral positions of the contralateral and the ipsilateral limbs, respectively. Thus, $I_{AB/AD} = 0$ if the lateral positions of the left and right limbs were symmetrical in relation to the trunk, $I_{AB/AD} > 0$ if the contralateral limb was more abducted than the ipsilateral one, and $I_{AB/AD} < 0$ if the ipsilateral limb was more abducted than the contralateral one.

To estimate the trunk orientation in the transverse plane stabilized on the tilting platform, the rear view was used. We measured the roll tilt angle of the trunk at two conditions: when the animal was standing on the platform tilted to the right and to the left (shown by black and gray in Fig. 3C). The average of these two angles was termed "the stabilized angle" [the angle between the vertical (green line) and the dashed red line in Fig. 3C]. With perfect stabilization of the dorsal side-up trunk orientation, the stabilized angle is equal to 0 (Fig. 3C, left panel), while positive and negative values of the stabilized angle indicate stabilization of orientation with the ipsilateral and contralateral roll tilt, respectively. To characterize the efficacy of postural corrections, we calculated the coefficient of postural stabilization $K_{STAB} = 1 - β/α$, where β is the amplitude of the trunk roll tilt, and α is the

amplitude of the platform tilt (Fig. 3I). With perfect stabilization, $K_{STAB} = 1$; with no stabilization, $K_{STAB} = 0$.

To characterize the trunk orientation in the transverse plane maintained during locomotion, the position of the spine in relation to the left-right body edges was characterized by using the top view. The trunk outline and its midline were drawn. Then the axis perpendicular to the midline at the level of hindlimbs (two-thirds of the distance from the nose to the base of the tail) was drawn. The midpoint of the body width was taken as "0". We measured the deviation of the spine (the red point in Fig. 4A–C) from the midpoint of the body width ("0") to the right (at the moment of the left hindlimb lift-off; $a$ in Fig. 4A) and to the left (at the moment of the right hindlimb lift-off) and calculated the average of these two values. This average was termed "the spine position during locomotion" and expressed in percent of the body half-width.

Asymmetry in abduction/adduction of the hindlimbs during locomotion was characterized by the abduction/adduction asymmetry index: $I_{AB/AD} = (b_{CONTRA} - b_{IPSI})/(b_{CONTRA} + b_{IPSI})$, where $b_{CONTRA}$ and $b_{IPSI}$ were the lateral positions of the ipsilateral and the contralateral limbs at the corresponding lift-off moments (when the maximal lateral displacement of the spine was observed, Fig. 4A). To assess possible asymmetry in temporal parameters of locomotor movements of the left and right limbs, first, the duration of the swing and stance were determined, as well as duty cycle was calculated. The period between the moments of the paw liftoff and touchdown was considered as the swing phase and the rest of the cycle as the stance phase. The duty cycle was defined as the duration of the stance phase divided by the total step cycle duration and represents the proportion of the step during which the limb is in ground contact. Then, to quantify the left-right asymmetry in locomotor movements, the ratio between values of stance duration, swing duration, and duty cycle of the contralateral and ipsilateral hindlimbs were calculated. A ratio of 1 indicates perfect left–right symmetry, while deviations from 1 reflect interlimb differences.

## Tissue immunochemistry
Mice were euthanized by anesthetic overdose with pentobarbital (250 mg/kg), and perfused transcardially with 4 °C saline followed by 4% paraformaldehyde. Brain and spinal cord tissue was dissected free and then postfixed in 4% paraformaldehyde for 3 h at 4 °C. Tissue was cryoprotected by incubation in 30% sucrose in phosphate-buffered saline (PBS) overnight. Tissue was then embedded in Neg-50 medium (Thermo Fisher Scientific) for cryostat sectioning. Coronal sections were obtained on a cryostat and mounted on Superfrost Plus slides (Thermo Fisher Scientific). Brainstem and spinal cord coronal sections were cut at 40 μm thickness.

For immunohistochemical detection of mCherry, sections were incubated overnight with a rabbit anti-DsRed/tdTomato/mCherry (1:1000, Clontech, catalog no. 632496). Both the primary and the secondary antibodies were diluted in 1% bovine serum albumin (BSA), 0.3% Triton X-100 in 0.1 M PBS. Following incubation with the primary antibody over night at 4 °C, the sections were incubated for 3 h with secondary antibody: Alexa-568 anti-rabbit (1:500; Jackson Immuoresearch, Cat# 111-585-003). Then, slides were washed in Phosphate buffered saline with tween (PBS-T, 0.5% triton), counterstained with Hoechst 33342 (1:2,000; Thermo Fisher Scientific, catalog no. 62249) and mounted with coverslips using glycerol containing 2.5% diazabicyclooctane (DABCO; Sigma-Aldrich). Sections were imaged using either a Zeiss widefield epifluorescence microscope or a Zeiss LSM 800 confocal microscope using Zen acquisition software.

## Heatmap generation
To build heatmaps showing the extent of the infected area in the brainstem and areas of the spinal cord gray matter with terminals of the infected neurons, we used images of coronal sections of the brainstem and spinal cord (cervical segments C4-C6, thoracic segments T5-T9, and lumbar segments L2-L5) containing the infected areas. Four borders of an image were aligned with the section edges. The image area was divided into 500 grid regions of interest (ROIs) by 25 columns and 20 rows. The fluorescence

intensity $F$ was measured in each ROI, the maximal $F$ across all 500 ROIs was found, and $F$ was normalized to this maximum. To obtain an averaged heatmap for a particular brainstem or spinal cord level, we used heatmaps built for individual sections taken at this particular level from different animals, and for each ROI we averaged the normalized F values across this set of heatmaps. The images were processed by using Image J and the heatmaps were generated by using MATLAB.

## RNAscope in situ hybridization
For in situ hybridization (RNAscope), the tissue sections were processed using RNAscope Multiplex Fluorescent v2 Assay (Advanced Cell Diagnostics, ACD, 323110) according to the manufacturer's instructions. In brief, the sections were air dried for 1 h at room temperature (RT) and placed in the PBS for 5 mins then dehydrated through sequential EtOH steps (50%, 70%, 100%, 100%) before air drying at RT. Then the sections were treated with hydrogen peroxide for 10 min at RT, washed in water, and treated with target retrieval reagents for 5 min at 99 °C. Sections were rinsed with water and treated with 100% EtOH for 3 min at RT and air dry again at RT. Tissues were then treated with protease III for 30 min at 40 °C before being rinsed with water. Specific probes were hybridized with sections for 2 h at 40 °C in a humidified oven, rinsed in wash buffer, and then stored overnight in 5× saline sodium citrate (SSC) buffer. The next day, the sections underwent a series of AMP1 ~ 3 incubation steps to amplify and develop the signals. The corresponding HRP channels and fluorophores (TSA Plus Fluorophores, Akoya) were applied to develop the signals from the hybridized probes. Specifically, the *Slc17a6* (*Vglut2*; #456751-C1) probe was used to label excitatory glutamatergic neurons, the *Slc32a1* (*Vgat*; #319191-C1) probe to label inhibitory GABAergic/glycinergic neurons, and the *CamkIIa* probe (#445231-C2) to detect CaMKIIa-expressing neurons. After finishing all the RNAscope steps, the sections were applied to the immunohistochemistry with the RFP antibody and Nissl.

## Statistics and reproducibility
Sample size was not estimated a priori to obtain a given power. Mice were randomly allocated to different groups for the in vivo experiments using a block design to limit confounders. A protocol was prepared before the study. This protocol was not registered. Data sampling and analysis were not blinded. All quantitative data in this study are presented as the mean ± SD. Mean values were calculated as averages of the mean from each animal. Each parameter was measured in an individual animal 8-20 times. The Student's $t$ test (two-tailed) was used to characterize statistical significance when comparing different means. No formal tests of assumptions were performed. Spearman's rank correlation coefficient was calculated to correlate the changes in the duration of the righting reflex movement with the asymmetry indexes. To evaluate the statistical significance of difference in relative numbers of glutamatergic and inhibitory CaMKIIa neurons in MdV-IRt area, we used Pearson's $\chi^2$ test. The significance level was set at $p = 0.05$.

## Reporting summary
Further information on research design is available in the Nature Portfolio Reporting Summary linked to this article.

## Data availability
The data supporting the findings of this study are available in Supplementary Data 1. Any other relevant information is available from the corresponding author upon reasonable request.

## Code availability
This study did not use any custom code. All analyses were performed with standard software as described in the Methods.

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

## Acknowledgements
This work was supported by grant from NIH (R01 NS-064964) to TGD; by grants from Swedish Research Council (2020-02502) to TGD; KI fonder (2024-2026) to LJH; Alice and Wallenberg foundation (KAW2022-0130) to FL; Olle Engkvist foundation (228-0277) to FL; NSTC postdoctoral research abroad program from Taiwan (112-2917-I-564-029) to SHC.

## Author contributions
P.V.Z.: Conceptualization, Investigation, Methodology, Writing–review and editing. V.F. L.: Investigation, Writing—review and editing. S.-H.C.: Methodology, Writing—review and editing. F.L.: Methodology, Funding acquisition, Writing—review and editing. T.G. D.: Conceptualization, Funding acquisition, Investigation, Methodology, Validation, Visualization, Writing—original draft. L.-J.H.: Conceptualization, Formal analysis, Investigation, Methodology, Validation, Visualization, Writing—original draft.

## Funding

## Competing interests
The authors declare no competing interests.
