## [Transparent Peer Review file · Communications Biology]

Role of CaMKIIa reticular neurons of caudal medulla in control of posture

Corresponding Author: Dr Li-Ju Hsu

Version 0:

Reviewer comments:

Reviewer #1

(Remarks to the Author)

The aim of this study by Zelenin and coll. was to identify the specific functions of a subset of reticular neurons, located in the ventral and intermediary parts of the medulla (MdV-IRt) and expressing CamKIIa. The authors used a series of behavioral tests in mice to investigate whether the activity level of such neurons might impact body orientation in space, and showed that activating or inactivating this specific population on one side disrupted body verticality (lateral tilt) and affected limb position relative to the vertebral axis, in both static and dynamic contexts. They additionally showed that CamKIIa neuronal population comprised both excitatory and inhibitory neurons, and that part of them were reticulospinal command neurons projecting all along the spinal cord. Based on these observations the authors state that MdV-IRt neurons are responsible for maintaining the body vertical posture (dorsal side up) in mice and that activity asymmetry between left and right-side populations of CamKIIa reticular neurons results in postural adjustment.

This study clearly provides new data in the field, the experiments reported look well designed, and the analysis is mostly well conducted. However, several conclusions seem overstated, and there are some flaws that need careful attention.

Major comments

The fact that unbalanced CamKIIa neurons activity distorted both a deviation of the body axis from the vertical and limb extension on one side makes the conclusion drawn by the authors that CamKIIa neurons are specifically involved in maintaining body verticality by acting on the axial motor system largely overstated. Indeed, it is not clear whether limb extension did not provoke the observed body axial deviation, the latter being only a passive mechanical result of the former. None of the experiments or analyses reported in the manuscript really addressed this point that is however mandatory to draw any conclusion about the spinal targets of MdV CamKIIa neurons activity.

Why were not all experimental conditions analyzed identically? For example, when the impact of modulating the activity of CamKIIa neurons out of the MdV was tested, only head and trunk tilt is analyzed (which showed no statistically significant effects), and the authors concluded that these neurons did not contribute to the control of posture. But what about limb positions? Responses to ground tilt? Dynamic postural control during locomotion? So, here again, conclusions were drawn too fast. Another example, when Vglut2 neurons impact was tested, head and trunk verticality was not analyzed (or at least, not reported in the manuscript); can't we imagine that the strong C shape evoked by Vglut2 neurons activation simply hides the discrete axial deviation observed during CamKIIa neurons manipulation? Here, maybe combining Vglut2 neurons activation and CamKIIa neurons inactivation would allow to identify each population specific role. Globally, it is difficult for the reader to compare between all experimental conditions when not the same parameters are systematically and similarly analyzed.

Conclusions on righting experiments (page 26) should be revised or, at least, discussed because the authors did not take into account the fact that the spatial arrangement of limbs was totally disturbed when CamKIIa neurons were manipulated: extension of the contralateral hindlimb definitively would impede, or even prevent (2 mice), righting in that direction... maybe not a specific effect of CamKIIa on the axial postural system, rather just a biomechanical impossibility.

Major statistical concerns

Page 21, top: using a t-test is absolutely not appropriate here because you compare proportions (%) of a whole; it requires the use of a K^2 test between your three populations of neurons, glutamatergic, inhibitory, and the remaining non-glu/non-inhib neurons (which may be interesting to illustrate also in the spatial distributions in fig9 F and G).

Page 35, first line: "(Each parameter was) measured in an individual animal 8-80 times": please be more specific when 8 or 80 repeated measures were made and if such a very different number of measures were made in different groups for a given

parameter: the variability for 80 measures will be 10 smaller than variability for 8, which would bias the results and, consequently, invalid any statistical comparison.

Other comments

Page 4, line 5: close bracket after Murray et al., 2018

Page 4, same paragraph: the link between the previous GVS experiments and the present study, which is evoked again in the discussion, is not obvious and lacks experiments demonstrating that these specific CamKIIa neurons in the MdV do integrate signals from vestibular afferents or central vestibular neurons (unless this is already known and references should be added)

Page 4, line 11: correct author name for Takakusaki

Page 7, fig2 legend, line 12-13: "In B,C,H..." sentence not easy to read. Consider rephrasing as "CamKIIa-RNs activation: B,C,H,J, n=7; CamKIIa-RNs inactivation: E,F,I,K, n=6"

Page 8, end of 2nd paragraph: although contralateral hindlimb extension is obvious in CamKIIa activated animals, there was no significant variations in ipsilateral limb position, so the assertion that ipsilateral were adducted should be removed

Page 9, line 15: change "addiction" for "adduction"

Page 16, last paragraph (and 1st sentence page 17): none of the effects of CamKIIa neurons inactivation were significant so it is not justified to conclude that "righting to the ipsilateral side was affected stronger"

Page 23, figure 10: please indicate what insets (D-F) are supposed to illustrate and use this data in the text

Page 24, last result paragraph: it would be very informative to know which proportion of MdV-IRt reticulospinal neurons are indeed CamKIIa neurons

Page 25, line 6: I don't understand the reference to fig11, since the sentence relates bibliographic data on GVS stimulation

Page 25, line 10: "... located in MdV-IRt area"

Page 26, last paragraph, 1st sentence: causality is not shown. In addition, as is, the sentence contradicts the later conclusion that CamKIIa neurons act on the axial postural system

Page 27, line 8 from bottom: "intensity of branching in the cervical region higher than in lumbar region"; however, the impact of manipulating CamKIIa neurons is much stronger on hindlimbs. Please discuss

Page 28, 2nd paragraph: this paragraph seems out of the study scope. Yes, CamKIIa are involved in plasticity but not only: also in synaptic structural homeostasis, pain sensitivity (neuropathic pain), neuron physiology (dysregulation involved in most, if not all, brain diseases) ... Consider removing this paragraph

Page 28, last discussion paragraph: there are several overstatements. Dorsal side up position is maintained in all animals, only a light deviation of the vertical axis is observed. In addition, the study does not demonstrate that left/right CamKIIa neurons activity becomes asymmetrical to trigger postural adjustments in compensation for ground surface lateral inclination (in contrast the authors showed that imposed asymmetry did not affect the postural response to ground inclination)

Page 29, line 20: it is indicated here that experiments started 3-6 weeks after virus injection: does it mean that experiments were performed on 11-18-week-old mice, or these 3-6 weeks were already taken into account in the "8-12 weeks old" indicated at the bottom of the preceding page?

Page 31, line 10: insert space between "skin" and "along"

Page 31, line 15: not clear: it is indicated here that locomotion was filmed from above. I thus suppose that experiments were made in a corridor rather than a tunnel (as indicated at bottom of page 30)

Page 33, line 7 from bottom: what is PB? I suppose it is PBS

Page 33, line 5 from bottom: define PBS-T, and indicate the detergent concentration used

Page 34, line 8: maybe precise which neuronal types are targeted with the probes used for RNAscope

Reviewer #2

(Remarks to the Author)

Zelenin et al. identify a specific class of reticular neurons (CaMKIIa in the caudal medulla) that are crucially involved in control of posture in mice. They show that unilateral activation of these neurons induces ipsilateral body tilt and limb asymmetry, which is actively stabilized during standing and walking, while inactivation produces opposite effects. Anatomical analysis revealed that CaMKIIa-RNs include a reticulospinal population projecting mainly ipsilaterally and are predominantly excitatory. These findings provide mechanistic insights into how supraspinal circuits modulate postural tone through asymmetric drive to the spinal cord. Importantly, they reveal a molecularly defined population within the reticular formation that maintains dorsal-side-up orientation, advancing our understanding of the neural basis of balance.

The major claims of the paper are original and well supported by the data. The identification of a functionally relevant, genetically defined reticular neuron population represents a novel and important contribution to the field of motor control and will be of interest not only to those studying posture and locomotion but also to the broader neuroscience community focused on supraspinal contributions to movement and neurological motor disorders. The statistical analyses appear appropriate, and the work is convincingly presented. While some additional clarification and reporting would enhance reproducibility, the experimental design is sound, and the conclusions are well justified.

Overall the paper presents very interesting and important results and I only have few minor comments:

1. From the paper, it is not clear why specifically this class of neurons was targeted. An explanation in the introduction (or discussion) would be very helpful.
2. It would be good include videos of the key tests and for the different animal groups.

3. Some measures that are important to understand the results are not explained in the main text, but the reader is referred to the methods. Effectively it isn't possible to understand the results without looking up the definitions in the methods. I suggest to provide short explanation of how to interpret these parameters in the results when they are first introduced. This includes "extension/flexion asymmetry index", "abduction/adduction asymmetry index", "Stabilized angle".
4. The results on maintained asymmetry during locomotion are very interesting. I was wondering if other parameters were recorded as well, such as swing/stance timings (which would allow to calculate duty factor asymmetry and phase differences), or limb kinematics? Even simply showing videos of locomotion would give the reader a much better idea of how the locomotor behavior looks like.
5. On page 7, the authors mention that CNO caused a gradually development of the asymmetric phenotype. Can this be reported more rigorously? How does the transition look like? Are there any qualitative changes?
6. Some discussion on the effect of the used chemogenetics would be helpful. Will activated neurons still be modulated by input or are they over-activated? What about inactivated neurons, can they still be engaged by natural input. I.e. should the effect be interpreted as modulatory or as overriding natural activity?
7. Is anything known about the inputs these neurons (or neurons with cell bodies in the same location) receive? Similarly, is anything known about projection targets to supraspinal motor centers?
8. Figures are sometimes difficult to interpret because the sign indicating the side of asymmetry (ipsilateral vs. contralateral) is inconsistent across measures—for some, ipsilateral values are shown as positive and contralateral as negative, while for others the opposite is true. Using a consistent convention throughout would improve clarity.
9. Figure 5D,E: are the data points with delta duration of infinity included in the correlation calculation? If so, they need to be removed.
10. Discussion about reorganization of reticulospinal neurons after spinal cord injury could potentially benefit from including Lemieux M, Karimi N, Bretzner F. Functional plasticity of glutamatergic neurons of medullary reticular nuclei after spinal cord injury in mice. Nature communications. 2024 Feb 20;15(1):1542.
11. Gi V2a neurons are mentioned in the introduction but it is not clear how they relate to the studied class. Some discussion might be interesting, since they result in extensor activation when evoking stops or turns. Has any postural effect of their manipulation been reported or excluded?
12. Page 4 & elsewhere: "the chemogenetic approach" -> "a chemogenetic approach"
13. Figure 4D, 5B,C only show differences to control values, but no absolute values are shown. For one, that's different to how other data is presented, but also potentially hides errors of analysis. Reporting of absolute values (including controls) would be preferable).
14. Page 9: "addiction" -> adduction
15. MdV-IRt wasn't defined at first use.
16. Figure 7F,G: without reading caption it isn't clear that percentage of each respective cell type is shown.
17. Figure 11: panel E missing although there is a panel F.
18. Figure 2: Sample shown in panel G doesn't seem to match data of panel H. In G the ipsilateral hindlimb is right of the midline (negative value) yet no data point with negative b_ipsi is shown in H.
19. This is a matter of style, but I was hoping for a bit more interpretation of the results when they are reported.
20. Number of male/female mice wasn't reported.
21. It is not clear to me if the control animals also received CNO or not.

Reviewer #3

(Remarks to the Author)

How neural networks originating in the brainstem and targeting neurons of the spinal cord are organized to control body position to maintain a balanced body position was examined in this study. The Authors utilized genetically-driven activation and inactivation of neural subpopulations within the brainstem. Designer receptors activated by designer drugs (DREADDs) were utilized to excite or inhibit brainstem neurons expressing CaMKII in wild-type mice.

The major claims of the paper are that the CaMKII α neurons of the brainstem drive postural control in mice. Unilateral activation of CaMKII α neurons evoked ipsilateral roll tilt of the head and trunk. This movement was defined as composite of flexion/adduction of the ipsilateral limbs and extension/abduction of the contralateral limbs. Interestingly, the body roll tilt was actively stabilized on the tilting platform and maintained during walking. Thus the CaMKII neurons were controlling dynamic

postural changes and not only static changes. Unilateral inactivation of CaMKIIa-RNs evoked the opposite effects. To verify that the CamKII-positive neurons have projections to influence the limbs and therefore being part of reticulospinal pathways, the Authors have performed histological analyses and found that in the caudal medulla, there are CaMKII-expressing reticulospinal neurons that project to the spinal cord mainly through ipsilateral lateral funiculus and terminate in the intermediate regions of the cord from cervical to lumbar levels.

These experiments are novel and unique. Elegantly designed work and analysis in terms of movement and also in terms of neuroanatomy which maximized the results obtained from the work. These results are interest to others in the community focused on brainstem and spinal neural control; motor control and sensory-motor integration. This work is of potential high interest to all those working on vestibular control of movement as well as interneural networks.

The work is convincing, and it will influence thinking in the field in terms of methodology. However, in order to improve the manuscript, there are a few issues that the Authors need to address.

- 1) In the Introduction, the statement "Although it is documented that neurons of the pontomedullary reticular formation contribute to control of posture," – requires a reference. Please, add a source for this statement.
- 2) Also in the Introduction, the Authors should include a summative, succinct overview of the "molecularly defined neural populations" within the brainstem and summarize the evidence leading to target the choice of neurons, the CamKII neurons in this study. The introduction currently suggests that there was a haphazard selection of the CamKII population to be examined; yet in the Discussion there are several points made.
- 3) Figure 1: a wide –field image showing the region of the brainstem from which this high-magnification image was taken is warranted to show that targeting the MdV is evident.
- 4) From Fig 1a, the claim that a "our findings reveal a more substantial presence of these cells in these two areas." is not supported. The claim of "more substantial presence" requires quantitative data. Was there any quantification performed to support this claim? Where are those data displayed/provided?
- 5) The rostro-caudal distribution of the injected viral constructs also need to be discussed. Please, comment on how extensive transfection was evident (as your injected systemic CNO to activate neurons and if neurons in an extensive rostro-caudal region were influenced; then weak effects could be explained differently than it is currently presented in the manuscript.
- 6) How did you assess the comparative weakness stated on page 9: "Also, abduction of the ipsilateral limbs and adduction of the contralateral limbs were much weaker expressed during unilateral CaMKIIa-RNs inactivation (Fig. 2I) as compared to those observed during unilateral activation (Fig. 2H).
- 7) The conclusion that " Thus, unilateral inactivation of CaMKIIa-RNs during locomotion evoked displacement of the spine toward the contralateral side suggesting that the trunk orientation with a contralateral roll tilt was maintained." - would require evidence of activity recorded from trunk muscles or kinematic assessment refined to testing trunk activation. The observations you report could also be explained by lack of excitation of trunk muscles. What evidence supports your conclusion- be more detailed when describing this.
- 8) Figure 4: How many steps were used to calculate angles? How long were the step cycles? Alteration of the cycles is expected with weaker trunk control and did that effect your calculations?
- 9) The appropriateness and validity of any statistical analysis has one potential problem. The small number of mice in two groups (VGat –cre and VGlut-cre n=3) is questionable with the stated parametric test used for comparisons, or the authors should provide more clear descriptions of the number of tests compared to justify the use of this test.
- 10) Some of the sentences in the Discussion should be checked for grammatical correctness.

Version 1:

Reviewer comments:

Reviewer #1

(Remarks to the Author)

Comments on the revised version of the manuscript by Zelenin and coll. about the characterization of a ventral medulla CamKIIa-expressing neuronal population, and its possible role in maintaining the verticality of the trunk ventro-dorsal axis in mice.

First of all, my apologies for my misunderstanding of the conclusions drawn in the original version and my own interpretation about axial system involvement. Second, the revised version includes several additional hints for the reader, which makes the manuscript really easier to follow. Nevertheless, I still have a few, mostly minor concerns.

Statistical concerns

Figure 9: I am not convinced by the authors justification, based on tests' assumptions, for the use of a t-test instead of chi-square test for comparing proportions of neurons: the only K_i^2 assumption is that values must be >5 ; if it is not the case, you can otherwise use a Fischer test. In contrary, t-test has much more assumptions (continuous variables, normality, variance equality...) which is not the case. Statistics on this specific point still seem not adapted (but it's minor, given the overall quality of the work).

General minor comment

I have a little problem with the "dorsal-side-up" notion used in the manuscript. In all experimental conditions except in righting experiments, mice are and remain dorsal-side-up: only a deviation from verticality of the body ventro-dorsal axis is observed. Maybe the report would gain clarity avoiding this ambiguous terminology.

Minor comments

Page 6, line 5-7, and Figure 1E: new images were added to show the rostral-caudal extent of the virus spreading, and they are given as representative of the animal sample. It seems on these pictures that virus largely spread in the inferior olive region (pictures 3 and 4), a region that may be involved in limb spatial configuration (see Funato et al 2021 who showed that IO lesions may trigger dystonia-like symptoms, including uncontrolled limb extension). I suggest this point to be considered in the discussion.

Page 6, line 17-19: it is now reported in the revised version that CNO effects started with only a head roll, followed by trunk roll and asymmetry in limb left-right organization. If so, could the head and then trunk tilts be attributed to limb asymmetry, as postulated by the authors? What was the delay between head, then trunk, then limb configuration changes? Please clarify this sequence of events and, perhaps, reconsider initial interpretations.

Page 6, line 26, Supplemental video 1: on the video, it seems that the left hindlimb finally lost contact with the ground, as if the mouse was slowly laying on its right side during CNO injection. Was it the case? Was it always observed under CNO activation?

Page 9, line 19: probably a mistake in the value reported for the control E/F index SEM (likely 0.02).

Page 13, line 18: what does this assumption suggest regarding CaMKIIa neurons function? At the end, are they involved in sensory-triggered postural correction, as it seems to be suggested later in the discussion?

Page 15, line 5-8: consider reformulating the sentence 'During inactivation the changes were significant (...)' since non-significant changes are also indicated onto the brackets.

Page 15, line 8 and elsewhere: change "insignificant" (which means negligible) for "non-" or "not significant".

Page 15, Figure 4: please give the graphs in F and G panels the same Y axis range, for better visual comparison.

Page 17, line 14 (figure 5 legend): I suppose it is a "Spearman's test".

Page 20, line 3-5: Were the animal samples identical for CaMKIIa neuron activation in figures 2, 6 and 8? I suppose it was (same number, same condition), but if so, why the reported CaMKIIa/MdV-IRt values are not the same between figure 2 and figures 6/8? Please clarify whether the group of 7 mice was used as control values.

Page 26, line 16: insert space between "the" and "induction".

Page 28-29: the discussion about the different effects of activation/inactivation at rest and during locomotion is very interesting. Maybe the authors should consider discussing also, in these contexts, the existence of two distinct CaMKIIa populations (inhibitory and excitatory) which specific activation or inactivation could have totally different impact according to the behavioral context.

Page 30, line 12: insert space between "the" and "yaw"

Page 31, line 5-7: V1 INs are mostly known to modulate locomotor pattern (structure, frequency; Gosnach et al, Falgairolle et al), which CaMKIIa activation/inactivation also does (change in cycle and stance duration). This reinforces the possibility that CaMKIIa neurons project on this specific IN population.

Page 31, line 10: suppress coma after "neurons".

Reviewer #2

(Remarks to the Author)

The revised manuscript represents a clear improvement; most of my comments and the comments of the other reviewers appear to have been addressed adequately. Needless to say, I'm still very enthusiastic about this work, and believe the results are convincing and important.

I have a few minor comments:

p13 lines 27ff, Fig 4: It was difficult for me to understand how exactly "lateral displacement of these spine oscillations" was quantified. A clearer definition with reference to the corresponding plots in the figure would be helpful.

p23 line 27, Fig 9C,D "rather similarly distributed" -> this should be quantified/tested. From the figure, it appears as if Vglut2+ neurons were more medial and dorsal than Vgat+. Also, for Fig 9C, D it would be better to show densities (e.g. as isopotential lines) rather than just individual dots.

reviewer 1 comment 17 and response: I believe that a brief statement about the possible involvement of propriospinal pathways (similar to the response to R1) in the discussion would be beneficial as otherwise the discrepancy between anatomy and behavior isn't addressed well.

p27 line 29-31: "However, the asymmetry in activity of CaMKIIa-RNs can be evoked by other types of sensory information, e.g., on the basis of the visual information about a laterally tilted surface in front of the animal." This statement could use references (what inputs do these regions receive), elaboration.

p29 line 9ff: Lack of asymmetry in timing of locomotor movements is mentioned but no data and/or analysis are shown. Thus it is impossible to judge what this statement means. I would like to suggest adding a figure for the locomotor timing/asymmetry data, including exemplary stance diagrams, even if there isn't a significant effect. The figure can also be in the supplementary material but is necessary.

Cycle, stance and swing durations are reported, including their changes with inactivation/activation of CaMKIIa-RNs. The changes with inactivation could be explained by a change in speed. Yet, I couldn't find any description of the effect of any manipulation on speed in the manuscript, except for a statement in the discussion that mentions a decrease of speed due (p 29, l14). The effect on speed should be reported and cycle, stance, swing duration comparisons should ideally be compared at the same speed or the effect of speed should be controlled for otherwise.

p31 lines 7ff: This should probably be a new paragraph. I suggest adding a sentence explaining why this distinction between different CaMKIIa RNs is discussed and why it wasn't addressed in the present paper. Merely for context.

p4 line 17-26: The rationale for targeting CaMKIIa still isn't clear to me from the added statement. Obviously, the presented results provide sufficient justification.

p6, line 21: should say "from time to time"

p26, line 16: would be helpful to have GVS spelled out again

p5 line 21: Acronyms MdV and IRt haven't been defined in the main text.

Reviewer #3

(Remarks to the Author)

The Authors have addressed all queries listed in the first round of review. The revised manuscript presents clear and significant findings that will advance the field of motor control research.

One small correction- pg 32, line 19-Viscotears is for lubrication of the eyes vs. dehydration. Otherwise, I have no further comments.

Version 2:

Reviewer comments:

Reviewer #1

(Remarks to the Author)

It seems to me that the authors have satisfied the recommendations made by the three referees. I think it clearly improved the manuscript and made it easier for readers. The study is now quite remarkable, and the paper of broad interest.

No more concerns. Congratulations to the authors for this very nice contribution.

Reviewer #2

(Remarks to the Author)

The authors have addressed all of my remaining minor comments and concerns, and, as far as I can tell, those raised by the other reviewer as well. I congratulate the authors on this nice piece of work.

Response to referees

Reviewer #1 (Remarks to the Author):

The aim of this study by Zelenin and coll. was to identify the specific functions of a subset of reticular neurons, located in the ventral and intermediary parts of the medulla (MdV-IRt) and expressing CamKIIa. The authors used a series of behavioral tests in mice to investigate whether the activity level of such neurons might impact body orientation in space, and showed that activating or inactivating this specific population on one side disrupted body verticality (lateral tilt) and affected limb position relative to the vertebral axis, in both static and dynamic contexts. They additionally showed that CamKIIa neuronal population comprised both excitatory and inhibitory neurons, and that part of them were reticulospinal command neurons projecting all along the spinal cord. Based on these observations the authors state that MdV-IRt neurons are responsible for maintaining the body vertical posture (dorsal side up) in mice and that activity asymmetry between left and right-side populations of CamKIIa reticular neurons results in postural adjustment. This study clearly provides new data in the field, the experiments reported look well designed, and the analysis is mostly well conducted. However, several conclusions seem overstated, and there are some flaws that need careful attention.

Major comments:

Comment 1. The fact that unbalanced CamKIIa neurons activity distorted both a deviation of the body axis from the vertical and limb extension on one side makes the conclusion drawn by the authors that *CamKIIa neurons are specifically involved in maintaining body verticality by acting on the axial motor system largely overstated*. Indeed, it is not clear whether limb extension did not provoke the observed body axial deviation, the latter being only a passive mechanical result of the former. None of the experiments or analyses reported in the manuscript really addressed this point that is however mandatory to draw any conclusion about the spinal targets of MdV CamKIIa neurons activity.

Response 1. We are confused by this comment. Our manuscript does not contain conclusion that *“CaMKIIa neurons are specifically involved in maintaining body verticality by acting on the axial motor system”*. We demonstrated that asymmetry in activity of CaMKIIa-RNs evoked left-right asymmetry of limbs (flexion/adduction of one limb and extension/abduction of another one) which most likely led to the body roll tilt. However, we agree with the Reviewer that the contribution of the axial muscles to control of the body orientation in the transverse plane cannot be ruled out. In the revised manuscript this is mentioned (P 29, L24-31).

As mentioned in Discussion, future experiments will clarify which of CaMKIIa sub-populations located in MdV-IRt area (interneurons, reticulospinal neurons or both) evoked the behavioral effects (P31, L7-9). In case CaMKIIa reticulospinal neurons contribute to control of the body roll tilt, the fact that one of the areas of arborization of their axons coincides with the area where motoneurons of axial muscles are located (laminae X) supports assumption about contribution of the axial muscles to muscles to execution of the body roll tilt. This issue is clarified in the revised manuscript (P31, L10-13).

Comment 2. **(i)** Why were not all experimental conditions analyzed identically? For example, when the impact of modulating the activity of CaMKIIa neurons out of the MdV was tested, only head and trunk tilt is analyzed (which showed no statistically significant effects), and the authors concluded that these neurons did not contribute to the control of posture. But what about limb positions? Responses to ground tilt? Dynamic postural control during locomotion? So, here again, conclusions were drawn too fast. **(ii)** Another example, when Vglut2 neurons impact was tested, head and trunk verticality was not analyzed (or at least, not reported in the manuscript); can't we imagine that the strong C shape evoked by Vglut2 neurons activation simply hides the discrete axial deviation observed during CaMKIIa neurons manipulation? Here, maybe combining Vglut2 neurons activation and CaMKIIa neurons inactivation would allow to identify each population specific role. Globally, it is difficult for the reader to compare between all experimental conditions when not the same parameters are systematically and similarly analyzed.

Response. **(i)** We apologize for insufficient clarity in description of aims of specific experiments that explain differences in analysis of their results.

In experiments with unilateral activation/inactivation of CaMKIIa-RNs located in MdV-IRt area, we subjected animals to a number of tests to demonstrate that the body roll tilt caused by asymmetry in CaMKIIa-RNs activity in animals standing on horizontal surface is actively stabilized (the test on the tilting platform) and maintained during locomotion. To clarify the role of limbs in induction of the body roll tilt in these tests, configuration as well as position of the limbs in relation to the trunk were analyzed.

The aim of experiments with unilateral activation of CaMKIIa-RNs in neighboring to MdV-IRt areas, was to clarify whether the effect of CaMKIIa-RNs on *the body orientation in the transverse plane* was specific for MdV-IRt area. Similarly, the aim of experiments with unilateral activation of GAD67 and Vglut2 neurons in MdV-IRt area was to clarify whether the effect on *the body orientation* is specific for CaMKIIa-RNs in this area. By other words whether unilateral activation of corresponding neurons evoked or did not evoke the body roll tilt. Since we demonstrated that asymmetry in activity of these neurons did not evoke the body roll tilt in standing animals, additional tests as well as analysis of limb configurations were not performed. We suggested that the abovementioned neurons do not contribute to control of the body orientation in the transverse plane. This is clarified in the revised manuscript (P18, L27; P19, L13; P20, L8). In the present study, we had no aim to investigate the functional role of these neurons in control of other aspects of posture or other motor behaviors. We agree with Reviewer that it will be interesting to reveal the possible role of GAD67 and non-CaMKIIa Vglut2 populations located in MdV-IRt, as well as CaMKIIa-RNs located outside of MdV-IRt area in control of other aspects of posture or other types of motor behaviors. However, this is a topic for future studies.

(ii) Unfortunately, it was impossible to analyze the head and trunk orientation during unilateral activation of Vglut2 neurons, since the animals continuously performed movements by the head and limbs accompanied by a strong trunk bending and circling. Since we demonstrated that the majority of CaMKIIa-RNs are Vglut2 neurons, the activated population most likely contained CaMKIIa-RNs. We agree with Reviewer that during activation of Vglut2 neurons the effect of CaMKIIa-RNs on the body orientation most likely was hidden by the strong behavioral effects of non-CaMKIIa Vglut2 neurons, which was completely different from that evoked by CaMKIIa-RNs (the body bending in the yaw plane). This is clarified in the revised manuscript (P30, L8-12). We are grateful to Reviewer for suggestion

of beautiful experiment for demonstration the behavioral effects of non-CaMKIIa Vglut2 neurons for our future studies.

Comment 3. Conclusions on righting experiments (page 26) should be revised or, at least, discussed because the authors did not take into account the fact that the spatial arrangement of limbs was totally disturbed when CaMKIIa neurons were manipulated: extension of the contralateral hindlimb definitively would impede, or even prevent (2 mice), righting in that direction... maybe not a specific effect of CaMKIIa on the axial postural system, rather just a biomechanical impossibility.

Response. We are confused by this comment, since we did not make conclusion that distortion of righting is a specific effect of CaMKIIa-RNs on the axial muscles. By contrast, we completely agree with Reviewer and concluded that "...asymmetry in configurations of the left and right limbs caused by unilateral activation/inactivation of CaMKIIa-RNs – extension and abduction of the hindlimb on the side of the less active subpopulation of CaMKIIa-RNs and simultaneous flexion and adduction of the hindlimb on the opposite side – distorted righting reflex toward the side with lower CaMKIIa-RN activity" (P18, L20-23).

Major statistical concerns:

Comment 4. (i) Page 21, top: using a t-test is absolutely not appropriate here because you compare proportions (%) of a whole; it requires the use of a χ^2 test between your three populations of neurons, glutamatergic, inhibitory, and the remaining non-glu/non-inhib neurons **(ii)** which may be interesting to illustrate also in the spatial distributions in fig9 F and G).

Response. (i) We appreciate the reviewer's concern about the appropriate statistical test. We compared the proportion of CaMKIIa neurons co-expressing Vglut2 or Vgat mRNA using six sections from three animals per group. Each data point reflected the percentage of double-labeled cells in a single section. Since the groups were from different animals and analyzed separately, the data do not meet the assumptions for a chi-square test. Instead, we used an independent t-test to compare mean proportions across sections. While this approach has limited power due to sample size, the difference was robust ($p = 0.017$).

(ii) The relative number of Vglut2+ and the relative number of Vgat+ neurons were calculated in different sections of individual animals. Thus, unfortunately, our data does not allow us to show the relative number of glutamatergic, inhibitory, and the remaining non-glu/non-inhib neurons at different medio-lateral and dorso-ventral levels of MdV-IRt area.

Comment 5. Page 35, first line: "(Each parameter was) measured in an individual animal 8-80 times": please be more specific when 8 or 80 repeated measures were made and if such a very different number of measures were made in different groups for a given parameter: the variability for 80 measures will be $\sqrt{10}$ smaller than variability for 8, which would bias the results and, consequently, invalid any statistical comparison.

Response. We apologize for the misprint. The actual number of repeated measurements per animal ranged from 8 to 20, depending on experimental conditions and video quality. The misprint has been corrected in the revised manuscript (P38, L21). While this variation affects the precision of the mean calculated within each animal (i.e., the standard error of the mean is smaller with more repetitions), it does not bias the group-level statistical comparisons, which were based on the mean value per animal. In our analysis, each animal contributed one averaged data point per condition. Therefore, variability in within-animal sample size only affects how precisely we estimate the mean value for each animal — not the validity of between-animal statistical tests.

Other comments

Comment 6. Page 4, line 5: close bracket after Murray et al., 2018

Response. In the revised manuscript, references are replaced by reference numbers.

Comment 7. Page 4, same paragraph: the link between the previous GVS experiments and the present study, which is evoked again in the discussion, is not obvious and lacks experiments demonstrating that these specific CaMKIIa neurons in the MdV do integrate signals from vestibular afferents or central vestibular neurons (unless this is already known and references should be added).

Response. We cannot agree with Reviewer that there is no link between previous GVS experiments and the present study. Left-right asymmetry in both vestibular inputs and CaMKIIa-RNs activity evokes change of the body orientation in the transverse plane that is actively stabilized indicating that both evoke shift of the set-point of the postural control system controlling the body orientation in the transverse plane. Our hypothesis is that the mechanisms behind a shift of the set-point caused by GVS and by asymmetry in CaMKIIa-RNs activity are similar. To increase clarity, the corresponding part of the Discussion was modified (P26, L15-17; P26, L1-14).

As noted in Introduction, since reticulospinal neurons (including those in the medulla) receive substantial vestibular input, one can assume that left/right asymmetry in activity of reticulospinal neurons contributes to the shift of the set-point of the postural system caused by GVS (P3, L32-34). Whether specifically CaMKIIa-RNs in the MdV-IRt area receive vestibular input and thus potentially can contribute to change of the body orientation caused by GVS remains unknown. However, the asymmetry in activity of CaMKIIa-RNs can be evoked on the basis of other types of sensory information, e.g., on the basis of the visual information about laterally tilted surface in front of the animal. Inputs from multiple upstream regions (including superior colliculi) to glutamatergic MdV neurons have been demonstrated (Esposito et al., 2014). This is clarified in the revised manuscript (P27, L25-33).

Comment 8. Page 4, line 11: correct author name for Takakusaki.

Response. We apologize for the misprint. In the text of the revised manuscript, references are replaced by reference numbers. In the reference list the misprint is corrected (P40, L33).

Comment 9. Page 7, fig2 legend, line 12-13: “In B,C,H...” sentence not easy to read. Consider rephrasing as “CamKIIa-RNs activation: B,C,H,J, n=7; CamKIIa-RNs inactivation: E,F,I,K, n=6”

Response. The sentence was modified according to Reviewer’s suggestion (P8, L6-7).

Comment 10. Page 8, end of 2nd paragraph: although contralateral hindlimb extension is obvious in CamKIIa activated animals, there was no significant variations in ipsilateral limb position, so the assertion that ipsilateral were adducted should be removed

Response. We apologize for this inaccuracy. In the revised manuscript, the sentence was corrected and only adduction of the forelimb (that was statistically significant) was mentioned (P9, L10-11).

Comment 11. Page 9, line 15: change “addiction” for “adduction”

Response. The misprint is corrected (P10, L2).

Comment 12. Page 16, last paragraph (and 1st sentence page 17): none of the effects of CamKIIa neurons inactivation were significant so it is not justified to conclude that “righting to the ipsilateral side was affected stronger”

Response. We thank the Reviewer for pointing this out. We agree that the original phrasing overstated the conclusion given that no statistically significant effects were observed. We have revised the relevant sentence in the Results section (P18, L5-7) to report the numerical trend without making interpretative claims, and to clearly state that the difference was insignificant.

Comment 13. Page 23, figure 10: please indicate what insets (D-F) are supposed to illustrate and use this data in the text

Response. We thank the Reviewer for pointing out the poor description of the Fig. 10. The Fig. 10 legend as well as corresponding text in the Results section were revised to clarify that D, E, and F panels shows position of the mCherry⁺ axons at the cervical, thoracic, and lumbar levels of the spinal cord, respectively (P25, L15-19; P26, L5-7).

Comment 14. Page 24, last result paragraph: it would be very informative to know which proportion of MdV-IRt reticulospinal neurons are indeed CamKIIa neurons

Response. We agree that knowledge about the proportion of MdV-IRt reticulospinal neurons that express CaMKIIa would be informative. This would require experiments involving dual

injections of Cre and Cre-dependent viral tracers, respectively, into the MdV-IRt and the spinal cord to selectively label and quantify this subpopulation. To reveal a relative number, differences in axonal projections as well as functional role of CaMKIIa and non-CaMKIIa reticulospinal neurons located in MdV-IRt area is a goal for future studies.

Comment 14. Page 25, line 6: I don't understand the reference to fig11, since the sentence relates bibliographic data on GVS stimulation

Response. We thank the reviewer for pointing this out. The corresponding part of the Discussion section was revised (P26, L15-17; P26, L1-14). Also, see *Response to Comment 7*.

Comment 15. Page 25, line 10: "... located in MdV-IRt area"

Response. In the revised manuscript, this part of the Discussion section was modified.

Comment 16. Page 26, last paragraph, 1st sentence: causality is not shown. In addition, as is, the sentence contradicts the later conclusion that CaMKIIa neurons act on the axial postural system

Response. As we already mentioned above (see *Response to Comment 1*), our manuscript did not contain conclusion that CaMKIIa-RNs act on the axial muscles.

In terrestrial quadrupeds, limbs play a crucial role in stabilization of the dorsal side-up body orientation in the transverse plane. As indicated in the manuscript (P10, L17-24), postural corrections aimed to restore the dorsal side-up body orientation distorted by the lateral tilt of the support, consist of extension of the limbs on the side of the tilt and simultaneous flexion of the opposite limbs. They are caused by an increase and decrease in activity of extensor muscles in the limbs on the side of the tilt and in the limbs on the opposite side, respectively. These limb movements lead to the body roll tilt (displacement of the dorso-ventral axis of the body toward the vertical) (Beloozerova et al., 2003; Deliagina et al., 2000, 2006). Asymmetry in activity of CaMKIIa-RNs caused simultaneous extension of limbs on one side, flexion of limbs on another side, and a change of the body orientation in the transversal plane (the roll tilt). This allowed us to suggest, that as in case of postural corrections, the body roll tilt was caused by asymmetry in configuration of the left and right limbs. However, the contribution of the trunk muscles to the execution of the body roll tilt cannot be ruled out. This is clarified in the revised manuscript (P29, L24-31).

Comment 17. Page 27, line 8 from bottom: "intensity of branching in the cervical region higher than in lumbar region "; however, the impact of manipulating CaMKIIa neurons is much stronger on hindlimbs. Please discuss

Response. As indicated in Discussion (P31, L7-9), it is not clear which sub-population of CaMKIIa neurons (reticulospinal, interneurons, or the entire population of CaMKIIa-RNs However,) produce the described behavioral effects. In case if reticulospinal CaMKIIa neurons play a crucial role in induction of the behavioral effects, one of possible explanations of strong effect on hindlimbs despite denser axonal branching of CaMKIIa reticulospinal

axons in cervical region, is that the effects are mediated by propriospinal interneurons receiving input from CaMKIIa reticulospinal neurons terminating in cervical and thoracic segments. However, since contribution of CaMKIIa reticulospinal neurons to observed behavioral effects is not demonstrated, we prefer to avoid the discussion of this issue in the manuscript.

Comment 18. Page 28, 2nd paragraph: this paragraph seems out of the study scope. Yes, CamKIIs are involved in plasticity but not only: also in synaptic structural homeostasis, pain sensitivity (neuropathic pain), neuron physiology (dysregulation involved in most, if not all, brain diseases) ... Consider removing this paragraph

Response. Our intention in this paragraph was not to suggest an exclusive role of CaMKIIa in synaptic plasticity, but rather to briefly highlight one specific and well-documented function—activity-dependent plasticity—that is particularly relevant in the context of post-injury reorganization of motor systems. A recent study (Lemieux et al., 2024) highlights functional plasticity in glutamatergic reticular neurons after spinal cord injury, further supporting the potential role of CaMKIIa-expressing MdV-IRt neurons in adaptive recovery processes. We have revised the paragraph to clarify its relevance (P31, L14-22) and prefer to keep it in the manuscript.

Comment 19. Page 28, last discussion paragraph: there are several overstatements. **(i)** Dorsal side up position is maintained in all animals, only a light deviation of the vertical axis is observed. **(ii)** In addition, the study does not demonstrate that left/right CamKIIa neurons activity becomes asymmetrical to trigger postural adjustments in compensation for ground surface lateral inclination (in contrast the authors showed that imposed asymmetry did not affect the postural response to ground inclination).

Response. **(i)** We do not quite understand this comment by Reviewer. Indeed, all terrestrial quadrupeds actively stabilize the dorsal side-up body orientation in different environmental conditions. However, to maintain it on the laterally tilted surface, the body roll tilt in relation to surface should be evoked. Our study showed that asymmetry in activity of CaMKIIa-RNs can evoke the body roll tilt. During unilateral activation of CaMKIIa-RNs its magnitude was substantial (in average approximately 30°).

(ii) The last paragraph (as well as entire manuscript) does not contain conclusion that CaMKIIa-RNs contribute to generation of postural limb reflexes (PLRs) that contribute to postural adjustments in response to lateral tilts. What was demonstrated, that left-right asymmetry in CaMKIIa-RNs (most likely tonic) activity caused by unilateral chemogenetic activation/inactivation changed the gains of postural limb reflexes generated by the left and right limbs that led to the body roll tilt. The efficacy of postural corrections caused by PLRs were not affected, just another body orientation was stabilized with the same precision. However, one cannot exclude that CaMKIIa-RNs contribute also to generation of PLRs. To clarify this issue is a question for future studies. This is mentioned in the revised manuscript (P27, L18-24).

Comment 20. Page 29, line 20: it is indicated here that experiments started 3-6 weeks after virus injection: does it mean that experiments were performed on 11-18-week-old mice, or these 3-6 weeks were already taken into account in the “8-12 weeks old” indicated at the bottom of the preceding page?

Response. We thank the reviewer for pointing out this unclarity. The age range “8–12 weeks” stated in the Methods section refers to the age of the mice at the time of injection. We have now clarified this explicitly in the revised Methods section to avoid confusion (P32, L6).

Comment 20. Page 31, line 10: insert space between “skin” and “along”

Response. Corrected (P34, L19).

Comment 21. Page 31, line 15: not clear: it is indicated here that locomotion was filmed from above. I thus suppose that experiments were made in a corridor rather than a tunnel (as indicated at bottom of page 30)

Response. Mice performed locomotion in a corridor. In the revised manuscript, “a tunnel” is substituted by “a corridor” (P34, L5-13)

Comment 22. Page 33, line 7 from bottom: what is PB? I suppose it is PBS

Response. The misprint is corrected (P37, L11).

Comment 23. Page 33, line 5 from bottom: define PBS-T, and indicate the detergent concentration used

Response. In the revised manuscript, PBS-T was defined and concentration indicated (P37, L11-12).

Comment 24. Page 34, line 8: maybe precise which neuronal types are targeted with the probes used for RNAscope

Response. According to suggestion of Reviewer, the neuronal identity targeted by each probe is specified in the Methods section of the revised manuscript (P38, L9-12).

Reviewer #2 (Remarks to the Author):

Zelenin et al. identify a specific class of reticular neurons (CaMKIIa in the caudal medulla)

that are crucially involved in control of posture in mice. They show that unilateral activation of these neurons induces ipsilateral body tilt and limb asymmetry, which is actively stabilized during standing and walking, while inactivation produces opposite effects. Anatomical analysis revealed that CaMKIIa-RNs include a reticulospinal population projecting mainly ipsilaterally and are predominantly excitatory. These findings provide mechanistic insights into how supraspinal circuits modulate postural tone through asymmetric drive to the spinal cord. Importantly, they reveal a molecularly defined population within the reticular formation that maintains dorsal-side-up orientation, advancing our understanding of the neural basis of balance.

The major claims of the paper are original and well supported by the data. The identification of a functionally relevant, genetically defined reticular neuron population represents a novel and important contribution to the field of motor control and will be of interest not only to those studying posture and locomotion but also to the broader neuroscience community focused on supraspinal contributions to movement and neurological motor disorders. The statistical analyses appear appropriate, and the work is convincingly presented. While some additional clarification and reporting would enhance reproducibility, the experimental design is sound, and the conclusions are well justified.

Overall the paper presents very interesting and important results and I only have few minor comments:

Comment 1. From the paper, it is not clear why specifically this class of neurons was targeted. An explanation in the introduction (or discussion) would be very helpful.

Response. According to suggestion of Reviewer, we clarified rationale for targeting CaMKIIa-expressing neurons in Introduction of the revised manuscript (P4, L17-26).

Comment 2. It would be good include videos of the key tests and for the different animal groups.

Response. According to suggestion of Reviewer, video recordings were added as supplementary material.

Comment 3. Some measures that are important to understand the results are not explained in the main text, but the reader is referred to the methods. Effectively it isn't possible to understand the results without looking up the definitions in the methods. I suggest to provide short explanation of how to interpret these parameters in the results when they are first introduced. This includes "extension/flexion asymmetry index", "abduction/adduction asymmetry index", "Stabilized angle".

Response. We thank the reviewer for this suggestion, which will help the reader to understand the findings. In the Results section of the revised manuscript, we explained how to interpret "extension/flexion asymmetry index" (P8, L13-15), "abduction/adduction asymmetry index" (P8, L25-27), "Stabilized angle" (P10, L26-28).

Comment 4. The results on maintained asymmetry during locomotion are very interesting. I was wondering if other parameters were recorded as well, such as swing/stance timings (which would allow to calculate duty factor asymmetry and phase differences), or limb kinematics? Even simply showing videos of locomotion would give the reader a much better idea of how the locomotor behavior looks like.

Response. Although limbs configuration and body orientation were altered during unilateral activation/inactivation of CaMKIIa-RNs, we did not find asymmetry in timing of locomotor movements performed by the left and right limbs. In the revised manuscript, we added panels F and G to Fig. 4 showing ratios between durations of the left and right limbs stance and swing durations, as well as duty cycles before and during activation (F) and inactivation (G) of CaMKIIa-RNs. No significant differences were observed between control and activation/inactivation conditions. The text describing Fig. 4F,G is added to the Results section (P14, L29-33; P15, L1-2).

Comment 5. On page 7, the authors mention that CNO caused a gradually development of the asymmetric phenotype. Can this be reported more rigorously? How does the transition look like? Are there any qualitative changes?

Response. In the revised manuscript, we described the time course and qualitative features of the CNO-induced phenotype in more detail (P6, L14-23). The asymmetry typically began to emerge in 15–20 minutes post-injection, initially as a subtle head roll tilt, followed by trunk roll tilt and left-right asymmetry in limb configurations. The phenotype reached maximal expression at approximately 40 minutes and remained stable for about 1–1.5 hours. Recovery was gradual, with a return to symmetrical posture in 3–4 hours after injection. These observations were consistent across animals.

Comment 6. Some discussion on the effect of the used chemogenetics would be helpful. Will activated neurons still be modulated by input or are they over-activated? What about inactivated neurons, can they still be engaged by natural input. I.e. should the effect be interpreted as modulatory or as overriding natural activity?

Response. Reviewer raised very important question, which unfortunately has no answer at present. While hM3Dq and hM4Di are designed to increase or reduce neuronal excitability, it remains unclear whether these neurons retain the ability to respond to inputs. To clarify this question is an important direction for future studies. This is noticed in the revised manuscript (P27, L21-24).

Comment 7. Is anything known about the inputs these neurons (or neurons with cell bodies in the same location) receive? Similarly, is anything known about projection targets to

supraspinal motor centers?

Response. In the present study, we did not map inputs to CaMKIIa-RNs in MdV-IRt area. However, previous studies demonstrated that neurons located in MdV receive inputs from several supraspinal motor centers. Specifically, Esposito et al. (2014, Nature) demonstrated that glutamatergic MdV neurons receive monosynaptic inputs from layer V pyramidal neurons in motor cortex, the superior colliculus, red nucleus, cerebellar deep nuclei, and other brainstem reticular nuclei. These findings suggest that MdV neurons are well-positioned to integrate converging motor-related signals from multiple upstream regions. This information is added to Discussion of the revised manuscript (P27, L30-32-21).

In relation to output targets of MdV neurons, it was shown that MdV contained reticulospinal neurons (Esposito et al., 2014; Xie et al., 2023) and we demonstrated that a part of them expressing CaMKIIa projected to all levels of the spinal cord. To our knowledge, the supraspinal targets of MdV neurons are unknown.

Comment 8. Figures are sometimes difficult to interpret because the sign indicating the side of asymmetry (ipsilateral vs. contralateral) is inconsistent across measures—for some, ipsilateral values are shown as positive and contralateral as negative, while for others the opposite is true. Using a consistent convention throughout would improve clarity.

Response. We thank Reviewer for the suggestion for improvement of clarity. We have carefully reviewed all figures and revised them to follow a consistent convention: positive and negative values now indicate contralateral and ipsilateral deviations, respectively. The following figures have been revised accordingly: Fig. 2B, E, H, I; Fig. 3D, E; Fig. 4D; Fig. 6G, H; and Fig. 8C, D.

Comment 9. Figure 5D,E: are the data points with delta duration of infinity included in the correlation calculation? If so, they need to be removed.

Response. To assess the relationship between asymmetry indexes and the duration of the righting reflex, we used Spearman's rank correlation rather than Pearson's correlation. Unlike Pearson's, which requires numeric values and assumes a linear relationship, Spearman's correlation evaluates monotonic relationships based on the rank order of the data. Since Spearman's method operates on ranks rather than raw values, data points with infinite values (e.g., animals that never completed the righting reflex) can still be included — they are simply assigned the highest rank (i.e., the longest duration of the reflex). This approach preserves their relative position in the dataset without distorting the correlation analysis. Therefore, we retained these data points in our analysis, and their inclusion is methodologically valid under the Spearman correlation framework.

Comment 10. Discussion about reorganization of reticulospinal neurons after spinal cord injury could potentially benefit from including Lemieux M, Karimi N, Bretzner F. Functional

plasticity of glutamatergic neurons of medullary reticular nuclei after spinal cord injury in mice. *Nature communications*. 2024 Feb 20;15(1):1542.

Response. We thank the reviewer for this suggestion that strengthening our assumption that glutamatergic CaMKIIa-RNs in MdV-IRt area may contribute to recovery of postural functions after spinal cord injury. In the revised manuscript, the study by Lemieux et al. (2024) is cited and our findings are placed in the context of plasticity of glutamatergic reticular neurons after spinal cord injury (P31, L17-19).

Comment 11. Gi V2a neurons are mentioned in the introduction but it is not clear how they relate to the studied class. Some discussion might be interesting, since they result in extensor activation when evoking stops or turns. Has any postural effect of their manipulation been reported or excluded?

Response. In introduction we just gave examples of specific motor effects evoked by different molecularly identified populations of neurons located in different reticular nuclei. We agree with Reviewer that effects revealed in our study and those produced by V2a neurons located in the gigantocellular reticular nucleus (Gi) should be discussed. The corresponding discussion is added to the revised manuscript (P30, L18-22). It was demonstrated that unilateral activation of V2a neurons in Gi induced ipsilateral trunk bending in standing mice and continuous ipsilateral turning during locomotion. In our study, a similar effect was observed during unilateral activation of Vglut2 RNs in MdV-IRt areas. Thus, glutamatergic neurons located in different reticular nuclei contribute to control of the body configuration in yaw plane. Unfortunately, in both studies, quantitative kinematic analysis illustrating contribution of limbs in induction of the body yaw turn was not performed.

Comment 12. Page 4 & elsewhere: "the chemogenetic approach" -> "a chemogenetic approach"

Response. Corrected (P4, L29; P32, L31).

Comment 13. Figure 4D, 5B,C only show differences to control values, but no absolute values are shown. For one, that's different to how other data is presented, but also potentially hides errors of analysis. Reporting of absolute values (including controls) would be preferable).

Response. According to the suggestion of Reviewer, Figures 4D and 5B,C are revised. Now they show values for both control and CNO conditions. The corresponding text in the Results section is modified accordingly (P13, L27-34; P14, L1-4; P18, L4-12).

Comment 14. Page 9: "addiction" -> adduction

Response. The misprint is corrected (P10, L2).

Comment 15. MdV-IRt wasn't defined at first use.

Response. In the revised manuscript, the abbreviations MdV and IRt are defined (P5, L22).

Comment 16. Figure 7F,G: without reading caption it isn't clear that percentage of each respective cell type is shown.

Response. We think that Reviewer's comment relates to Fig. 9F,G (since Fig. 7F,G does not contain percentages). In the revised manuscript, the corresponding text in the Results section was modified, to clarify that the values represent the percentage of Vglut2⁺ and percentage of Vgat⁺ CaMKIIa-RNs in MdV-IRt area at different medio-lateral (F) and dorso-ventral (G) levels (P23, L24-28).

Comment 17. Figure 11: panel E missing although there is a panel F.

Response. The misprint is corrected.

Comment 18. Figure 2: Sample shown in panel G doesn't seem to match data of panel H. In G the ipsilateral hindlimb is right of the midline (negative value) yet no data point with negative b_ipsi is shown in H.

Response. The panel H shows an example of limb configurations during unilateral activation of CaMKIIa-RNs. In the panel G, each point represents the average value of multiple trials performed by one animal. Since the value in G reflects trial-averaged measurements, it may differ from single-trial value observed in the same animal.

Comment 19. This is a matter of style, but I was hoping for a bit more interpretation of the results when they are reported.

Response. We intentionally chose a more descriptive and structured reporting style to maintain clarity and avoid conflating observations with interpretations. This approach allows readers to distinguish the raw findings from the interpretations and suggestions, which are considered in detail in the Discussion section. We believe that such separation strengthens the logical flow and transparency of the manuscript.

Comment 20. Number of male/female mice wasn't reported.

Response. In the revised manuscript, this information is provided in the Methods section (P32, L3).

Comment 21. It is not clear to me if the control animals also received CNO or not.

Response. In the present study, the control data were obtained in the same animals before CNO injection. This is indicated on P6, L10-13. Injection of CNO in animals without virus injection does not evoke any specific behavioral changes.

Reviewer #3 (Remarks to the Author):

How neural networks originating in the brainstem and targeting neurons of the spinal cord are organized to control body position to maintain a balanced body position was examined in this study. The Authors utilized genetically-driven activation and inactivation of neural subpopulations within the brainstem. Designer receptors activated by designer drugs (DREADDs) were utilized to excite or inhibit brainstem neurons expressing CaMKII in wild-type mice.

The major claims of the paper are that the CaMKII α neurons of the brainstem drive postural control in mice. Unilateral activation of CaMKII α neurons evoked ipsilateral roll tilt of the head and trunk. This movement was defined as composite of flexion/adduction of the ipsilateral limbs and extension/abduction of the contralateral limbs. Interestingly, the body roll tilt was actively stabilized on the tilting platform and maintained during walking. Thus the CaMKII neurons were controlling dynamic postural changes and not only static changes. Unilateral inactivation of CaMKII α -RNs evoked the opposite effects.

To verify that the CaMKII-positive neurons have projections to influence the limbs and therefore being part of reticulospinal pathways, the Authors have performed histological analyses and found that in the caudal medulla, there are CaMKII-expressing reticulospinal neurons that project to the spinal cord mainly through ipsilateral lateral funiculus and terminate in the intermediate regions of the cord from cervical to lumbar levels.

These experiments are novel and unique. Elegantly designed work and analysis in terms of movement and also in terms of neuroanatomy which maximized the results obtained from the work. These results are interest to others in the community focused on brainstem and spinal neural control; motor control and sensory-motor integration. This work is of potential high interest to all those working on vestibular control of movement as well as interneural networks.

The work is convincing, and it will influence thinking in the field in terms of methodology. However, in order to improve the manuscript, there are a few issues that the Authors need to address.

Comment 1. In the Introduction, the statement “Although it is documented that neurons of the pontomedullary reticular formation contribute to control of posture,” – requires a reference. Please, add a source for this statement.

Response. In the revised manuscript, the corresponding reference is added (Ref 24, P4, L7).

Comment 2. Also in the Introduction, the Authors should include a summative, succinct overview of the “molecularly defined neural populations” within the brainstem and summarize the evidence leading to target the choice of neurons, the CamKII neurons in this study. The introduction currently suggests that there was a haphazard selection of the CamKII population to be examined; yet in the Discussion there are several points made.

Response. We agree with Reviewer that in Introduction of the previous version of the manuscript we did not adequately explain the rationale behind targeting CaMKIIa-expressing neurons in the brainstem. In Introduction of the revised manuscript, we added a paragraph explaining the reasons for targeting this population (P4, L17-26).

Since our study is focused on functional role of specific molecularly identified neurons of reticular formation (CaMKIIa-RNs), we prefer to overview in Introduction only those molecularly identified neuronal populations of reticular neurons which functional role in control of specific aspects of motor behavior was demonstrated (P4, L11-16).

Comment 3. Figure 1: a wide –field image showing the region of the brainstem from which this high-magnification image was taken is warranted to show that targeting the MdV is evident.

Response. We thank the reviewer for the suggestion. In the revised manuscript, we added to Fig. 1B (a confocal image acquired at 10X magnification) a yellow rectangle to indicate the specific region shown at higher magnification (20X) in Fig. 1D. We have also adjusted the orientation of Fig. 1D to match the spatial layout of Fig.1B for clarity. The legend to Fig.1 was modified correspondingly (P5, L9-10).

Comment 4. From Fig 1a, the claim that a “our findings reveal a more substantial presence of these cells in these two areas.” is not supported. The claim of “more substantial presence” requires quantitative data. Was there any quantification performed to support this claim? Where are those data displayed/provided?

Response. We agree with Reviewer. In the revised manuscript, this statement was removed.

Comment 5. The rostro-caudal distribution of the injected viral constructs also need to be discussed. Please, comment on how extensive transfection was evident (as your injected systemic CNO to activate neurons and if neurons in an extensive rostro-caudal region were influenced; then weak effects could be explained differently than it is currently presented in the manuscript.

Response. We thank Reviewer for raising this issue. We used the same viral titer and injection volume in all experiments, including those targeting different anatomical sites. In the revised manuscript, this is indicated in Methods (P32, L26; P32, L34; P33, L1; P33, L8-10) and Results (P5, L19-20; P18, L30-32) sections. Thus, the differences in behavioral outcomes across groups are unlikely caused by differences in viral spread. This is mentioned in the Discussion of the revised manuscript (P29, L34; P30, L1-3).

For targeting CaMKIIa-RNs in MdV-IRt area, we used the same viral titer, injection volume, and injection coordinates in all animals which led to similar rostro-caudal, dorso-ventral and medio-lateral extension of infected neurons in individual animals. A representative example of the spatial distribution of transfected neurons was added to Fig. 1 (panel E) of the revised manuscript. The corresponding description of Fig. 1E is provided in its figure legend and Results section (P5, L12-15; P6, L5-7).

Comment 6. How did you assess the comparative weakness stated on page 9: “Also, abduction of the ipsilateral limbs and adduction of the contralateral limbs were much weaker expressed during unilateral CaMKIIa-RNs inactivation (Fig. 2I) as compared to those observed during unilateral activation (Fig. 2H).

Response. We thank Reviewer for pointing on the absence of statistical analysis supporting our conclusion. In the revised manuscript, we compared changes in abduction of the forelimb and hindlimb as compared to control, as well as changes in adduction of the forelimb and hindlimb as compared to control under two conditions. We found that the changes in abduction of both forelimb and hindlimb as compared to control were significantly larger during unilateral activation than during unilateral inactivation of CaMKIIa-RNs ($p = 0.0087$ and $p = 0.034$, respectively), while corresponding changes in adduction were insignificant. This information is added to the revised manuscript (P9, L22-31).

Comment 7. The conclusion that “ Thus, unilateral inactivation of CaMKIIa-RNs during locomotion evoked displacement of the spine toward the contralateral side suggesting that the trunk orientation with a contralateral roll tilt was maintained. (Page 15, Line 18-20)” - would require evidence of activity recorded from trunk muscles or kinematic assessment refined to testing trunk activation. The observations you report could also be explained by lack of excitation of trunk muscles. What evidence supports your conclusion- be more detailed when describing this.

Response. We thank Reviewer for highlighting this unclarity in our description. Observed during unilateral activation/inactivation of CaMKIIa-RNs displacement of the spine toward the edge of the body outline on the top view, indicates that the body orientation was maintained with some roll tilt. We clarified this in the revised manuscript (P13, L30-31). Most likely, as during standing, the body roll tilt during locomotion was caused by asymmetry in configurations of the left and right limbs performing stepping. One of the limbs performed stance at abducted and strongly extended configuration while another one at adducted and less extended (flexed) configuration. However, as in the case of the body roll tilt observed in standing animal, one cannot exclude that asymmetry in activity of trunk muscles twisting forequarter as well as those twisting hindquarters may contribute to maintenance of the body roll tilt during unilateral activation/inactivation of the CaMKIIa-RNs. This is discussed in the revised manuscript (P29, L24-31). See also *Response to Comment 16 by Reviewer 1*.

Comment 8. Figure 4: How many steps were used to calculate angles? How long were the step cycles? Alteration of the cycles is expected with weaker trunk control and did that effect your calculations?

Response. For calculation of the spine displacement as well as the abduction/adduction asymmetry index, we used about 8-10 locomotor cycles in each animal (N=5, 40 and 43 locomotor cycles, respectively, for control and activation conditions; N=4, 47 and 41 locomotor cycles, respectively, for control and inactivation conditions). This information is added to Fig. 4 legend.

We did not find asymmetry in locomotor movements of the left and right limbs (Fig. 4F,G) during activation as well as inactivation of CaMKIIa-RNs. However, during inactivation, we found a significantly increase in the cycle duration due to significant increase in the duration of stance. A decrease in the locomotor speed is unlikely to affect body orientation and limbs position in relation to trunk. The reason for the decrease in locomotor speed is not clear. It could be an effect of asymmetrical activity of CaMKIIa-RNs produced on rhythm generating part of CPG as well as a result of postural asymmetry making it difficult to maintain balance during locomotion on horizontal surface. In the revised manuscript, the abovementioned information as well as possible reasons for a decrease of locomotor speed during unilateral activation/inactivation of CaMKIIa-RNs are added (P14, L29-34; P15, L1-10; P30, L18-22).

Comment 9. The appropriateness and validity of any statistical analysis has one potential problem. The small number of mice in two groups (VGat-cre and VGlut-cre n=3) is questionable with the stated parametric test used for comparisons, or the authors should provide more clear descriptions of the number of tests compared to justify the use of this test.

Response. We thank Reviewer for pointing out the limitations of performing statistical comparisons with small sample sizes (n = 3 for Vgat-Cre group). We agree that the power of parametric tests such as the t-test is limited under these conditions, and that non-significant results in this context do not provide strong evidence for a true absence of effect. In light of this, we removed the statistical comparison of the GAD67 group (Vgat-Cre) against zero. Instead, 95% confidence interval for the change of parameters from control are indicated (P22, L12-18). We did not perform any statistical analysis for VGlut-cre mice.

Comment 10. Some of the sentences in the Discussion should be checked for grammatical correctness.

Response. According to suggestion of Reviewer, the grammar in the Discussion section was corrected by native English-speaking person.

Response to comments

Reviewer 1

Statistical concerns

Comment. Figure 9: I am not convinced by the authors justification, based on tests' assumptions, for the use of a t-test instead of chi-square test for comparing proportions of neurons: the only K_i^2 assumption is that values must be >5 ; if it is not the case, you can otherwise use a Fischer test. In contrary, t-test has much more assumptions (continuous variables, normality, variance equality...) which is not the case. Statistics on this specific point still seem not adapted (but it's minor, given the overall quality of the work).

Response. According to suggestion of Reviewer, Fig. 9E was modified and χ^2 test was applied to estimate statistical significance. Corresponding changes were made in Results (P23, L24) and Methods (P38, L25-26) sections

General minor comment

Comment. I have a little problem with the “dorsal-side-up” notion used in the manuscript. In all experimental conditions except in righting experiments, mice are and remain dorsal-side-up: only a deviation from verticality of the body ventro-dorsal axis is observed. Maybe the report would gain clarity avoiding this ambiguous terminology.

Response. We are grateful to Reviewer for the note that the term “the dorsal-side-up body orientation” was not defined. We prefer to keep this term in the revised manuscript. To avoid confusion, we gave a definition of this term in Abstract and in Introduction sections (P2, L1-2; P3, L1-2).

Minor comments

Comment 1. Page 6, line 5-7, and Figure 1E: new images were added to show the rostro-caudal extent of the virus spreading, and they are given as representative of the animal sample. It seems on these pictures that virus largely spread in the inferior olive region (pictures 3 and 4), a region that may be involved in limb spatial configuration (see Funato et al 2021 who showed that IO lesions may trigger dystonia-like symptoms, including uncontrolled limb extension). I suggest this point to be considered in the discussion.

Response. Indeed, in our study, some limited viral infection of the IO was observed due to its close proximity to the MdV. Therefore, we cannot exclude that our chemogenetic manipulations may affect IO neurons. However, we do not think that changes in activity of IO was the major source of the observed behavioral effects, since animals with weak IO infection sometimes displayed robust body roll tilt, whereas animals with strong IO infection could show relatively small tilt. Thus, most likely, involvement of IO in induction of behavioral effects was minimal. This is mentioned in the revised manuscript, as well as suggested by Reviewer study is cited (P29, L18-25).

Comment 2. Page 6, line 17-19: it is now reported in the revised version that CNO effects started with only a head roll, followed by trunk roll and asymmetry in limb left-right organization. If so, could the head and then trunk tilts be attributed to limb asymmetry, as postulated by the authors? What was the delay between head, then trunk, then limb

configuration changes? Please clarify this sequence of events and, perhaps, reconsider initial interpretations.

Response. The present study was not aimed to investigate the development in time effects of CNO injection and thus, unfortunately, we cannot present any quantitative data. We apologize for the bad formulation of our observations of CNO effects development that created confusion. The problem is that to notice a small roll tilt of the head is much easier than to notice a small roll tilt of the trunk. Thus, without measurements we cannot be sure that there is a delay between expression the head and trunk roll tilt. To avoid confusion and speculation, in the revised manuscript, we just indicated that effect was developed gradually and was maximally expressed in 40 minutes after CNO injection (P6, L21-22).

Comment 3. Page 6, line 26, Supplemental video 1: on the video, it seems that the left hindlimb finally lost contact with the ground, as if the mouse was slowly laying on it right side during CNO injection. Was it the case? Was it always observed under CNO activation?

Response. The video shows an example of very strong effect that was observed only in a few (N=2) mice. To avoid confusion, in the revised manuscript, the supplementary video 1 was substituted by the video of more representative example.

Comment 4. Page 9, line 19: probably a mistake in the value reported for the control E/F index SEM (likely 0.02).

Response. We apologize for the misprint. It is corrected in the revised manuscript (P9, L21)

Comment 5. (i) Page 13, line 18: what does this assumption suggests regarding CamKIIa neurons function? (ii) At the end, are they involved in sensory-triggered postural correction, as it seems to be suggested later in the discussion?

Response. (i) As indicated on Page 13, line 18, we demonstrated that asymmetry in activity of CaMKIIa-RNs does not affect the efficacy of postural corrections. It means, that postural corrections allow the animals to stabilize the dorso-ventral axis of the trunk on the tilting platform in control and after CNO with similar precision. However, in control, they stabilize the orientation of the dorso-ventral axis of the trunk close to vertical, while after CNO injection they stabilize orientation of the dorso-ventral axis of the trunk tilted toward the dominant side. (ii) As indicated in Discussion, to clarify whether left-right asymmetry in the tonic activity of CaMKIIa-RNs changes the gain of postural limb reflexes (PLRs) that leads to stabilization of a new orientation in the transverse plane, or whether PLRs in mice are mediated by CaMKIIa-RNs, is a question for future studies (P26, L34-35; P27, L1-2).

Comment 6. Page 15, line 5-8: consider reformulating the sentence ‘During inactivation the changes were significant (...’ since non-significant changes are also indicated onto the brackets.

Response. In the revised manuscript, the sentence was reformulated (P16, L15-19).

Comment 7. Page 15, line 8 and elsewhere: change “insignificant” (which means negligible) for “non-“ or “not significant”.

Response. “Insignificant” replaced by “non-significant” throughout the text (P16, L18; P16, L20; P9, L31; P27, L29).

Comment 8. Page 15, Figure 4: please give the graphs in F and G panels the same Y axis range, for better visual comparison.

Response. As suggested Reviewer, Fig. 4F,G was modified (P15).

Comment 9. Page 17, line 14 (figure 5 legend): I suppose it is a “Spearman’s test”.

Response. The misprint was corrected (P18, L7)

Comment 10. Page 20, line 3-5: Were the animal samples identical for CaMKIIa neuron activation in figures 2, 6 and 8? I suppose it was (same number, same condition), but if so, why the reported CamKIIa/MdV-IRt values are not the same between figure 2 and figures 6/8? Please clarify whether the group of 7 mice was used as control values.

Response. In Figs. 2B, 6G,H, and 8C,D, the data obtained in the same animals and at the same conditions are presented. However, in Fig. 2B, the mean values of the head roll tilt in control and after CNO application are shown, while in Figs. 6G,H and 8C,D a difference between two conditions (CNO and Control) are shown. That is why the values shown in Fig. 2B differs from those shown in Figs. 6G,H and 8C,D. The values shown in Fig. 6G,H and in Fig. 8C,D are the same.

Comment 11. Page 26, line 16: insert space between “the” and “induction”.

Response. Corrected (P26, L16)

Comment 12. Page 28-29: the discussion about the different effects of activation/inactivation at rest and during locomotion is very interesting. Maybe the authors should consider discussing also, in these contexts, the existence of two distinct CaMKIIa populations (inhibitory and excitatory) which specific activation or inactivation could have totally different impact according to the behavioral context.

Response. As we mentioned in the Discussion section, to reveal the functional role of excitatory and inhibitory subpopulations of CaMKIIa-RNs is a topic for future studies (P30, L16-19). In the present study, we activated/inactivated both populations simultaneously. Since the majority of CaMKIIa-RNs in MdV-IRt area are excitatory, one can assume that the observed behavioral effects are caused by excitatory sub-population. However, since we have no direct evidences, we prefer to avoid speculations about the possible roles of excitatory and inhibitory subpopulations of CaMKIIa-RNs in observed behavioral changes.

Comment 13. Page 30, line 12: insert space between “the” and “yaw”

Response. Corrected (P30, L11)

Comment 14. Page 31, line 5-7: V1 INs are mostly known to modulate locomotor pattern (structure, frequency; Gosnach et al, Falgairolle et al), which CaMKIIa activation/inactivation also does (change in cycle and stance duration). This reinforces the possibility that CaMKIIa neurons project on this specific IN population.

Response. We are thankful to Reviewer for this comment and add this assumption to the Discussion of the revised manuscript (P30, L30-33).

Comment 15. Page 31, line 10: suppress coma after “neurons”.

Response. Comma is removed (P31, L4)

Reviewer 2

Minor comments

Comment 1. p13 lines 27ff, Fig 4: It was difficult for me to understand how exactly "lateral displacement of these spine oscillations" was quantified. A clearer definition with reference to the corresponding plots in the figure would be helpful.

Response. The detailed explanation of how the spine position during locomotion was defined and quantified is provided in Methods section (P36, L6-15). We prefer do not overload Results section with analysis details and provide reference to Method section.

Comment 2. p23 line 27, Fig 9C,D "rather similarly distributed" -> this should be quantified/tested. From the figure, it appears as if Vglut2+ neurons were more medial and dorsal than Vgat+. Also, for Fig 9C, D it would be better to show densities (e.g. as isopotential lines) rather than just individual dots.

Response. According to suggestions by Reviewer, Fig. 9C,D was modified and densities of Vglut2+ and Vgat+ neurons in different zones of MdV-IRt area were compared. (P23, L24-27).

Comment 3. reviewer 1 comment 17 and response: I believe that a brief statement about the possible involvement of propriospinal pathways (similar to the response to R1) in the discussion would be beneficial as otherwise the discrepancy between anatomy and behavior isn't addressed well.

Response. According to suggestion by Reviewer, the statement about the possible involvement of propriospinal pathways is added to Discussion section. (P31, L4-8)

Comment 4. p27 line 29-31: "However, the asymmetry in activity of CaMKIIa-RNs can be evoked by other types of sensory information, e.g., on the basis of the visual information about a laterally tilted surface in front of the animal." This statement could use references (what inputs do these regions receive), elaboration.

Response. We are confused by this comment, since information about inputs to MdV-IRt area as well as corresponding reference is provided. In particular, the input from superior colliculi mediating visual information was indicated (P27, L12-14).

Comment 5. p29 line 9ff: Lack of asymmetry in timing of locomotor movements is mentioned but no data and/or analysis are shown. Thus it is impossible to judge what this statement means. I would like to suggest adding a figure for the locomotor timing/asymmetry data, including exemplary stance diagrams, even if there isn't a significant effect. The figure can also be in the supplementary material but is necessary.

Response. The lack of asymmetry in timing of locomotor movements during unilateral activation and inactivation of CaMKIIA-RNs is documented in Fig.4, F and G, respectively and described in the Results section (P14, L32-34; P15, L1-5).

Comment 6. Cycle, stance and swing durations are reported, including their changes with inactivation/activation of CaMKIIA-RNs. The changes with inactivation could be explained by a change in speed. Yet, I couldn't find any description of the effect of any manipulation on speed in the manuscript, except for a statement in the discussion that mentions a decrease of speed due (p 29, 114). The effect on speed should be reported and cycle, stance, swing duration comparisons should ideally be compared at the same speed or the effect of speed should be controlled for otherwise.

Response. The aim of the present study was to reveal the effects of CaMKIIa-RNs located in MdV-IRt area in control of posture. That is why the experiments were focused on changes in postural aspects of locomotion and thus, detailed study of the effects on the correlation of the speed with the structure of locomotor cycle was not performed. In both conditions (control

and activation/inactivation) animals freely selected the speed of walking. According to request by Reviewer, we added the comparison of the speed values in control and during unilateral activation/inactivation of CaMKIIa-RNs to Result section of the revised manuscript (P16, L13-22).

Comment 7. p31 lines 7ff: This should probably be a new paragraph. I suggest adding a sentence explaining why this distinction between different CaMKIIa RNs is discussed and why it wasn't addressed in the present paper. Merely for context.

Response. According to suggestion of Reviewer, in the revised manuscript, the discussion about the possible role of different subpopulations of CaMKIIa neurons located in MdV-IRt area is provided in a separate paragraph. (P31, L1-11)

Comment 8. p4 line 17-26: The rationale for targeting CaMKIIa still isn't clear to me from the added statement. Obviously, the presented results provide sufficient justification

Response. Our Lab is studying organization and operation of postural networks as well as changes in these networks underlying recovery of postural functions after incomplete spinal cord injury. As indicated in Discussion (P31, L12-20), since it is documented that CaMKIIa neurons contribute to plastic changes, one can expect that they may contribute to recovery of postural functions after spinal cord injury. As the first step, the present study was aimed to clarify whether CaMKIIa-RNs contribute to control of posture. We prefer to avoid such long description of the rationale in Introduction.

Comment 9. p6, line 21: should say "from time to time"

Response. Corrected as suggested by Reviewer (P6, L23).

Comment 10. p26, line 16: would be helpful to have GVS spelled out again

Response. According to suggestion by Reviewer, the abbreviation GVS was defined again in Discussion section. (P26, L15)

Comment 11. p5 line 21: Acronyms MdV and IRt haven't been defined in the main text.

Response. In the revised manuscript, abbreviations MdV and IRt are defined. (P6, L1-2; P6, L4).

Reviewer 3

Comment. One small correction- pg 32, line 19-Viscotears is for lubrication of the eyes vs. dehydration.

Response. The sentence was corrected according to suggestion by Reviewer: "Viscotears was used for lubrication of the eyes to prevent dehydration". (P32, L17)